# Mapping the temporal transcriptomic signature of a viral pathogen through CAGE and nanopore sequencing

**Dóra Tombácz, Balázs Kakuk, Gábor Torma, Ádám Fülöp, Ákos Dörmő, Gábor Gulyás, Zsolt Csabai, Zsolt Boldogkői** [ORCID]*

Department of Medical Biology, Albert Szent-Györgyi Medical School, University of Szeged, Szeged, Hungary

* boldogkoi.zsolt@med.u-szeged.hu

## Abstract

### Introduction

Equid alphaherpesvirus 1 (EHV-1), a veterinary pathogen belonging to the *Varicellovirus* genus, is responsible for significant economic losses in the global equine sector. This research involved timescale gene expression profiling and transcriptional reannotation of this herpesvirus.

### Methods

We employed CAGE sequencing on the Illumina platform to determine transcript start sites, alongside long-read direct cDNA sequencing on Oxford Nanopore Technology platform to detect full-length viral transcripts. Samples were collected in triplicate at nine distinct stages of the viral lifecycle. We also applied protein synthesis inhibition to determine the immediate-early gene expression of the virus. Earlier data on native RNA sequencing was also utilized to validate the results. The sequencing data were processed using the LoRTIA and NAGATA software tools.

### Results

The time-course analysis of viral transcript expression using long-read dcDNA-Seq enabled the characterization of these transcripts based on their kinetic behavior throughout the replication cycle. Furthermore, the study involved a comprehensive reannotation of the EHV-1 transcriptome. CAGE analysis helped identify the transcription start sites and promoter regions, while direct cDNA sequencing provided a more accurate approach to capturing full-length transcripts and isoform diversity. Through an integrated approach, we identified and validated numerous novel transcripts, thereby refining the EHV-1 transcriptome annotation. These methods allowed for a more detailed and accurate mapping of the EHV-1 transcriptome, uncovering previously unknown transcripts and refining the existing annotations.

**Data availability statement:** The sequencing datasets generated in this study are available at the European Nucleotide Archive under the accession: PRJEB52190 and PRJEB6233. The R codes used to perform the analysis and generate the plots are available at: https://github.com/Balays/EHV-1-dynamic.

**Funding:** National Research, Development and Innovation Office grant: K 142674 (ZB) and FK 142676 (DT).

**Competing interests:** The authors have declared that no competing interests exist.

## Conclusions

The shifting patterns in transcript isoforms and overlaps suggest a sophisticated regulatory network that enables EHV-1 to precisely modulate gene expression throughout its replication cycle. The presence of multiple isoforms per gene indicates that the virus can adapt to different stages of infection by producing a variety of transcripts. This likely enhances its genomic efficiency and allows it to respond more effectively to the host's environment.

## Introduction

Equid alphaherpesvirus 1 (EHV-1), also referred to as *Varicellovirus equidalpha1* [1] commonly presents with symptoms such as upper respiratory tract disease, spontaneous abortion in pregnant mares, neonatal death, and life-threatening myeloencephalopathy [2]. EHV-1 contains an approximately 150 kilobase pair double-stranded DNA genome. This genome is organized into two segments designated as unique short (US) and unique long (UL) regions, both surrounded by inverted repeats (IRs) [3,4]. The complete viral genome contains 80 open reading frames (ORFs) [5] among which five genes (ORF1, 2, 67, 71, and 75) are absent in other alphaherpesviruses with annotated genomes [6]. Given that four ORFs are situated in the IR region, the EHV-1 genome comprises a total of 76 unique protein-coding genes. We note that the term 'ORF' is used to denote the entire genes in EHV-1, not just the protein-coding parts. Similar to other alphaherpesviruses, EHV-1 can either productively infect cells or enter a latent state in specific peripheral neurons [7].

EHV-1 genes fall into three categories: immediate-early (IE), early (E), and late (L). The regulation of these genes is governed by viral transcription factors, specifically ORF5, 12, 63, 64, and 65 [8,9]. Notably, EHV-1 has a sole IE gene [10], ORF64, which bears homology to *rs1* gene of herpes simplex virus type 1 (HSV-1) coding for the ICP4 protein. Most of the viral E genes encode enzymes required for DNA replication, whereas the L genes specify structural proteins found in the virion, such as capsid and spike proteins. Late genes are further subdivided into leaky late (L1) and true late (L2), depending on their reliance on DNA replication for their expression [10].

Long-read sequencing (LRS), developed by Pacific Biosciences (PacBio) through synthesis-based sequencing and by Oxford Nanopore Technologies (ONT) through nanopore sequencing, has become essential in modern transcriptome analysis. The long-read RNA sequencing (lrRNA-Seq) technique is particularly effective in identifying transcription start sites (TSSs), transcription end sites (TESs), splice sites, alternative splicing, embedded transcripts, multigenic RNA molecules, and transcriptional overlap [11]. While lrRNA-Seq platforms can deliver full-length cDNA or native RNA sequences, they come with reduced throughput and a higher incidence of sequencing errors compared to short-read sequencing (SRS) platforms [12–17]. In transcriptomics, inaccuracies in sequencing are not a significant concern when the genomic sequence of a specific organism is well-established. The lrRNA-Seq technology based on ONT is highly suitable for native RNA sequencing. It guarantees correct read orientation and is free from the artefacts generated by reverse transcription and PCR [11,18], and it also facilitates the detection of RNA modifications [19].

Transcriptomic studies in herpesviruses have applied both PacBio and ONT approaches [15,17,20–26]. Temporal dynamics of herpesvirus gene expression has also been examined using SRS [27]. Our previous transcriptome profiling study, which employed a direct RNA sequencing (dRNA-Seq), identified the canonical EHV-1 transcripts, including mRNAs,

non-coding RNAs (ncRNAs) and many long multi-gene transcripts [28]. However, native RNA sequencing has limitations in accurately identifying TSSs due to 5'-truncation caused by motor protein stalling during sequencing. To address this and refine the EHV-1 transcriptome annotation, we integrated cap analysis of gene expression sequencing (CAGE-Seq) with direct cDNA sequencing (dcDNA-Seq). CAGE-Seq provides high-resolution TSS mapping, while dcDNA-Seq captures full-length transcripts without the 5'-truncation issues of dRNA-Seq. This combined approach enabled us to identify novel transcript isoforms and validate previously identified RNA molecules with greater accuracy. Using dcDNA-Seq, we determined the sequences of 27 samples collected at nine time points spanning 1–48 hours post-infection (hpi), with three replicates per time point. This extensive temporal sampling captured the full dynamics of viral gene expression across the complete infection cycle. By analyzing this time-resolved data, we clustered genes into *de novo* kinetic classes based on their expression dynamics.

## Methods

### Cells and viruses

In this study, we utilized the field isolate equid alphaherpesvirus 1 strain MdBio (EHV-1-MdBio), which was originally isolated from the organs of an aborted colt fetus in the 1980s at Marócpuszta, Hungary, and has been previously described [28]. The virus was propagated in a confluent rabbit kidney (RK-13) epithelial cell line (ECACC: 00021715). Cells were cultivated in DMEM (Sigma), supplemented with 10% fetal calf serum and 80 µg of gentamycin per ml (Gibco) at 37 °C in the presence of 5% $CO_2$. For the preparation of virus stock solution, cells were infected with 0.1 multiplicity of infection [MOI = plaque-forming units/cell]. Viral infection was allowed to progress until complete cytopathic effect was observed. As a next step, three successive cycles of freezing and thawing of infected cells were carried out to release of viruses from the cells. For the sequencing experiments, RK-13 cells were infected with 4 MOI of EHV-1-MdBio in three technical replicates. Infected cells were incubated for 1 h at 4 °C, followed by removal of the virus suspension and washing the cells with phosphate-buffered saline. As a next step, new culture medium was added to the infected cells, which were incubated for 1, 2, 4, 6, 8, 12, 18, 24, or 48 h. After the incubation, the culture medium was removed, and the infected cells were frozen at −80 °C until further use. Ethics approval is "Not Applicable," as no animal experiments were performed.

### Cycloheximide treatment of cells

RK-13 cells were grown in DMEM supplemented with 10% fetal bovine serum until they reached 60–70% confluency. The medium was then replaced with 5 mL of serum-free DMEM containing either 20 or 100 µg/mL cycloheximide (CHX). After a 1-hour incubation, this medium was substituted with 2 mL of a 10 MOI virus solution, also containing the same CHX concentrations, and the cells were incubated for either 6 or 8 hours. Post-incubation, the CHX-treated cells were washed once with PBS, scraped off the dish, and centrifuged at 2000 g for 2 minutes. Following the removal of the supernatant, the cell samples were immediately placed on dry ice for future analysis.

### RNA extraction

The RNA extraction was conducted using the NucleoSpin RNA kit from Macherey-Nagel. The process began by lysing the cells in a buffer containing chaotropic ions to deactivate RNases. This step facilitated the binding of DNA and RNA molecules to the silica membrane.

To eliminate any residual genomic DNA (gDNA), all samples were treated with DNase I. The total RNA was then eluted in nuclease-free water. Further purification to remove any remaining gDNA was achieved using the TURBO DNA-free™ Kit from Invitrogen. The RNA concentration in the samples was determined using the Qubit 4.0 fluorometer and the Qubit Broad Range RNA Assay Kit, also from Invitrogen (S1 Table). Quality control was performed with the Agilent TapeStation 4150, and only samples with RIN scores equal to or greater than 9.2 were used for cDNA synthesis and subsequent experiments.

## Purification of polyadenylated RNA

The poly(A)$^+$ RNA fraction was extracted from the total RNA using the Oligotex mRNA Mini Kit by Qiagen. Initially, the volume of each sample was adjusted to 250 μL with RNase-free water. Then, 15 μL of Oligotex suspension and 250 μL of OBB buffer, both from the Qiagen kit, were added to the samples. The mixture was heated to 70°C for 3 minutes and subsequently cooled to 25°C for 10 minutes. After centrifuging at 14,000×g for 2 minutes, the supernatants were discarded. The samples were then washed with 400 μL of OW2 wash buffer from the kit and transferred to spin columns provided in the kit, followed by centrifugation at 14,000×g for 1 minute. This washing step was repeated. Finally, the polyadenylated RNA was eluted from the membrane using 50 μl of pre-heated elution buffer from the Qiagen kit, collected in 60 μl elution buffer, with a second elution step performed to maximize the yield (S2 Table).

## RNA quantification

For measuring total RNA, we used the Qubit RNA BR Assay Kit from Invitrogen (Carlsbad, CA, United States). To quantify the poly(A)+ fraction, the Qubit RNA HS Assay Kit, also from Invitrogen (Carlsbad, CA, United States), was employed. The final concentrations of these RNA samples were determined using the Qubit® 4 fluorometer.

## cDNA quantification and quality assessment

The concentrations of cDNA samples and sequencing-ready libraries were ascertained using the Qubit dsDNA HS Assay Kit from Invitrogen (Carlsbad, CA, United States). The quality of RNA, crucial for sequencing, was evaluated using the Agilent 2100 Bioanalyzer for PacBio sequencing, and the Agilent 4150 TapeStation System for ONT sequencing. Samples with RIN scores of 9.6 or higher were selected for cDNA synthesis.

## CAGE sequencing

The CAGE-Seq protocol has been previously described [29]. Briefly, using the CAGE™ Preparation Kit (DNAFORM, Japan), we performed CAGE-Seq on viral genomic regions employing three biological replicates. Initially, 5 μg of total RNA and the kit's RT primer were mixed and heated at 65 °C. SuperScript III Reverse Transcriptase (Invitrogen) and a trehalose/sorbitol mixture (from the kit) were used for first-strand cDNA synthesis, followed by oxidation of the Cap's diol groups and biotinylation. RNase I (from the kit) digested single-strand RNA. Biotinylated samples were then bound to Streptavidin beads, washed, and cDNAs were released and purified. RNase mixture treated the samples to digest any residual RNA. Streptavidin beads, coated with tRNA, were prepared for linker ligation. After reducing the sample volumes using the miVac DUO Centrifugal Concentrator (Genevac), 5′- and 3′-linkers were ligated, followed by Shrimp Alkaline Phosphatase and USER enzyme treatments. The second cDNA strand was synthesized, treated with Exonuclease I, and samples were dried and resuspended in nuclease-free water. Single-stranded cDNA concentrations were measured using

Qubit 2.0 and the Qubit ssDNA HS Assay Kit. Purification steps employed RNAClean XP and AmpureXP Beads at various stages. Pooled libraries with different barcodes were sequenced on a MiSeq instrument using v3 (150 cycles) and v2 (300 cycles) chemistries (Illumina). The final concentration and quality of the libraries were assessed using Qubit 4.0 with a 1X dsDNA High Sensitivity (HS) Assay and TapeStation, respectively.

## Library construction and cDNA sequencing

Libraries for direct cDNA sequencing on the ONT MinION device were constructed using poly(A)$^+$-enriched samples. We followed the protocol of the ONT Direct cDNA Sequencing Kit (SQK-DCS109), as outlined in the kit's manual. Initially, RNA samples were mixed with VN primer (from the ONT kit) and 10 mM dNTPs, and heated at 65°C for 5 minutes. This was followed by the addition of 5x RT Buffer, RNaseOUT (from Thermo Fisher Scientific), and Strand-Switching Primer (SSP; from the ONT Kit), and a subsequent 2-minute heating at 42°C. The first cDNA strand synthesis involved the Maxima H Minus Reverse Transcriptase enzyme (from Thermo Fisher Scientific), with the reaction occurring at 42°C for 90 minutes, and enzyme inactivation at 85°C for 5 minutes. RNA strands from RNA-cDNA hybrids were removed using the RNase Cocktail Enzyme Mix (from Thermo Fisher Scientific) at 37°C for 10 minutes. The second cDNA strand was synthesized using LongAmp Taq Master Mix [from New England Biolabs (NEB)] and PR2 Primer (PR2P), with PCR reaction specifics is described in [28]. DNA fragments were then processed for end-repair and dA-tailing using the NEBNext End repair/dA-tailing Module (NEB) at 20°C for 5 minutes, followed by 65°C for 5 minutes. This step was followed by adapter ligation using the NEB Blunt/TA Ligase Master Mix (NEB) at room temperature for 10 minutes. The dcDNA-Seq libraries were barcoded as outlined in [28], and as per the ONT Native Barcoding (12) Kit instructions. The prepared cDNA libraries (200 fmol/flow cell) were purified and loaded onto ONT R9.4.1 SpotON Flow Cells, using a total of five flow cells for sequencing. To prevent "barcode hopping," samples from earlier and later time points were sequenced separately. After each enzymatic step, AMPure XP Beads were used for purification. The samples were then eluted in UltraPure™ nuclease-free water (from Invitrogen), and their concentration was measured using the Qubit 4.0 fluorometer and Qubit dsDNA HS Assay kit.

## Pre-processing and Data Analysis

**dcDNA sequencing data pre-processing.** The raw current signals obtained from MinION sequencing were basecalled to nucleotides with the Dorado-0.7.2 basecaller (https://github.com/nanoporetech/dorado/) using a quality threshold of 7. The resulting reads were aligned to the reference genome (accession number: NC_001491.2) using the minimap2 [30] program. During the alignment with minimap2, the following settings were applied: -ax splice -Y -C5 -cs. To identify TSS, TES, and intron positions, we used the LoRTIA toolkit (https://github.com/zsolt-balazs/LoRTIA). For evaluating dcDNA-Seq, the following settings were applied in the LoRTIA package: −5 TGCCATTAGGCCGGG –five_score 16 –check_in_soft 15–3 AAAAAAAAAAAAAAA –three_score 16 –s Poisson –f true. A transcript was accepted when its 5'- and 3'-adapters were accurate, and in the case of 3'-ends, false priming and template switching during intron identification were excluded. For introns, we accepted those annotated in dRNA sequencing for direct cDNA samples.

**dcDNA sequencing data processing.** For further analysis, we used an in-house developed R pipeline. Briefly, the "*stranded_only.bam*" files from the LoRTIA output were imported into the R environment using Rsmatools [31]. A database was then built from it, containing the count of unique mapping positions and the information from the bam-files regarding

LoRTIA's adapter searching using the data.table [32] R package [33]. This collection of unique mapping positions at the nucleotide level, considering both exons and introns along with their associated statistics, was henceforth termed "*transfrags*." These transfrags were used in the downstream pipeline as queries for reference transcript counting and for merging with the CAGE-Seq data. However, they are not true transcripts, as they can differ by as little as a single nucleotide. Because of this, they could also be used to calculate per-nucleotide and windowed coverage values, as well as 3'- and 5'-end counts.

**CAGE-Seq pre-processing and analysis.** The STAR aligner (version 2.7.3 a) [34] was used to map the reads to the EHV-1 reference genome (NC_001491.2), utilizing --genomeSAindexNbases 8 and default parameters. "Bam" files obtained from CAGE-seq were converted to BigWig format to detect 5'-end coverage. The CAGEfightR [35] package was used to determine TSS positions. We ran CAGEfightR with the following parameters: clusterUnidirectionally (pooledCutoff = 1, mergeDist = 10). This means that TSS positions within a 10-nucleotide window were clustered together (mergeDist = 10), and TSS clusters with a minimum pooled value (pooledCutoff = 1) of less than 1 were excluded from further analysis.

**Reference transcript counting.** We used the collection of annotated, filtered, and validated transcripts from our previous study as a reference collection, and the list of assembled transfrags (along with their per-sample counts, as described above) as queries to count the reference transcripts (from [28]) in each dcDNA sample using the GFF-compare tool [36]. However, since this tool tends to assign shorter transcript isoforms – those contained within another transcript - to the longer one, we ran this tool iteratively for each reference transcript separately. The results were then merged, and the distances between the transfrags (queries) and all their hits (reference transcripts) were calculated. The best hit (i.e., the closest reference transcript with the smallest distance) for each query (transfrag) was then selected as the transfrag's corresponding transcript. For counting reference isoforms, only hits classified as "equal to reference" (=) were retained, with a distance cut-off of 10 nucleotides for both ends and a 2-nucleotide wobble in intron positions to account for mapping issues. The R-packages rtracklayer [37] was used to export and import.gff3 files.

**TSS clusters validation.** Next, we assigned confidence levels to the TSS clusters (from the CAGEfightR output) by empirically combining support thresholds with quartile-based score bins. Clusters with support ≤ 2 and scores in the bottom quartile (Q1) were classified as low confidence (labelled with an asterisk [*]), while those with support of 3–5 and scores in Q2 or Q3 were assigned moderate confidence (labelled with two asterisks [**]). Clusters with support exceeding 5 or scores in the top quartile (Q4) were categorized as high confidence (labelled with three asterisks [***]). Clusters meeting only one of the high-confidence criteria were also classified as moderate confidence. This classification is heuristic rather than a formal statistical test, relying on observed distribution patterns to label CAGE TSS clusters as low, moderate, or high confidence.

**Transcript merging and TSS refinement.** To integrate the CAGE-Seq and dcDNA-Seq datasets, we merged CAGE TSS clusters with transfrags obtained from dcDNA-Seq. These transfrags contained reference transcript identities from the GFF-compare results and were used to refine TSS annotation. Only transfrags with a "correct" 5'-end (i.e., where the soft-clipped region contained the adapter sequence) were used.

The process consisted of two key steps: assigning CAGE TSS clusters to transfrags, and refining broad CAGE TSS clusters based on dcDNA-Seq 5'-ends.

1) Merging CAGE TSS clusters with transfrags

We first merged CAGE TSS clusters (from the CAGEfightR output) with transfrags (from dcDNA-Seq) based on their relative genomic positions. Specifically, a CAGE TSS cluster was

assigned to a transfrag if its 5′-end fell within the cluster boundaries. However, since some CAGE TSS clusters spanned up to ~150 nucleotides, additional refinement was necessary.

2) Refinement of TSS clusters using peak analysis

To improve the resolution of TSS positions, we refined each CAGE TSS cluster by leveraging dcDNA-Seq 5′-end counts. We applied a custom peak-analysis algorithm to each cluster, using dcDNA-Seq 5′-end read counts as weights. The refinement process was as follows:

a. Grouping adjacent positions into sub-clusters if: their genomic distance was ≤ 5 bp, and the cluster had not exceeded a maximum size of 25 positions.

b. Excluding positions with zero coverage to eliminate noise.

This refinement ensured that local hotspots of TSS usage were captured with high precision, producing compact and biologically meaningful TSSs, termed validated TSS peaks, for downstream analysis.

**Transcript assembly and validation.** Transcripts were reconstructed from the dcDNA dataset by pairing validated TSS peaks (dcDNA 5′-end peaks within the CAGE TSS clusters) with their corresponding TESs from the reference transcript list. For this, the following stringent criteria were used: (i) transfrags were annotated to a novel TSS only if their 5′-ends were within ±10 nucleotides of the refined TSS cluster, and (ii) their 3′-ends overlapped a known TES (also within ±10 nucleotides). This approach enabled the integration of CAGE-Seq and dcDNA-Seq datasets to annotate TSSs from transfrags that could not be assigned to a reference transcript but exhibited significant TSS activity based on both dcDNA and CAGE data. To annotate transcripts, we paired the refined (validated) TSS peaks with TESs from the reference transcript list, ensuring that (i) the transcript's 5′-end was within ±10 nucleotides of the refined TSS cluster, and (ii) the corresponding 3′-end overlapped a known TES within ±10 nucleotides. Each transcript required strict criteria for annotation:

(1) At least three dcDNA-Seq reads sharing the same TSS and TES coordinates,

(2) The presence of correct 5′-adapter sequences,

(3) Alignment with CAGE-Seq–derived TSS clusters and previously validated TESs.

Newly assembled transcripts were integrated with our prior annotation [28], allowing us to reintroduce previously excluded transcripts that now met refined criteria, as well as to add novel transcripts not previously detected.

We further evaluated our previous dRNA-Seq dataset using the NAGATA software [38] to validate the novel TSSs (and introns) identified by our LoRTIA and CAGEfightR-based workflow. We used NAGATA with the following settings: -m 1 -tg 2, with all other parameters set to default. This configuration was chosen to annotate rare TSSs, and introns in the dRNA-Seq data, which were confirmed by the dcDNA-Seq data exhibiting significantly higher read counts. This approach enabled the integration of CAGE-Seq and dcDNA-Seq datasets to annotate TSSs from transfrags that could not be assigned to a reference transcript but still exhibited significant TSS activity based on both dcDNA and CAGE data.

**Transcript classification.** The transcripts were classified based on their structural and functional features, including coding capacity (e.g., putative mRNAs, non-coding RNAs), monocistronic or multicistronic, and variations in untranslated regions (e.g., shorter or longer isoforms), compared to the canonical transcript.

(1) Putative embedded mRNAs: these transcripts contain a 5′-truncated ORF with an in-frame ATG and share a 3′-coterminus with the canonical ORF of the given gene.

(2) Non-coding RNAs: transcripts that lack ORFs.

(3) 5'-UTR isoforms: these can be mono- or polycistronic, containing the same ORF as the canonical transcript but differing in the length of their 5′-untranslated regions (UTRs).

(4) Antisense RNAs: transcripts that are complementary to and transcribed in the opposite direction of the genes.

**Filtering 5′-truncated ORF-carrying transcripts.** To further filter the TSSs of transcripts containing 5′-truncated ORFs with in-frame ATGs (putative mRNAs), the identified TSSs had to meet two criteria: (i) they had to appear in the NAGATA output within a 25 nucleotide wobble range (to account for potentially missing 5′-ends), and (ii) they required validation by CAGE based on their signal strength (5′-end read count) relative to the canonical TSS of their parent gene. Specifically, for a 5′-truncated isoform to be included, its TSS had to exhibit at least 5% expression compared to its canonical transcript.

*De novo Clustering of TSSs, TESs and Transcripts based on Dynamic Expression Patterns*

To identify groups of TSSs, TESs, and transcripts (TSS~TES) with similar temporal expression patterns, we performed de novo clustering on the viral read count-normalized canonical TSSs, TESs, and transcript expression data, respectively. In this context, "canonical" refers to the main (most abundant) feature—TSS or TES—for each gene. That is, we did not include short, long, or other transcript isoforms in this analysis.

For TSS clustering, we used reads from the time-resolved dcDNA-Seq dataset that (i) had a correct 5′-adapter, (ii) aligned with the canonical TSS, and (iii) had the correct 5′-end. For TES clustering, we used reads that (i) had a correct polyA-tail and (ii) whose 3′-ends overlapped with the canonical TES. For canonical transcript clustering, only reads that met the above criteria for both the TSS and TES were included. For each analysis, we used viral read count-normalized values, which were calculated differently for each sample by dividing each canonical TSS, TES, or transcript count by the total TSS, TES, or transcript read count, respectively. Only reads that fulfilled the above criteria were used as the basis for normalization.

The hierarchical clustering was conducted using the pvclust R package [39], which provides hierarchical clustering and bootstrap resampling for cluster validation. The complete linkage method and an uncentered correlation distance measure were applied. Cluster stability and significance were assessed using 1,000 bootstrap iterations, focusing on approximately unbiased (AU) p-values provided by pvclust. After generating a dendrogram, we evaluated cluster solutions ranging from 4 to 15 clusters, assessing their quality based on AU values and within-cluster sum of squares (WSS). Partitioning the data into 12 clusters struck a meaningful balance between resolution and interpretability. The hierarchical tree was cut into 12 clusters, each representing a distinct temporal expression pattern over the time course. This method was applied to all three data types: canonical TSSs, TESs and transcripts. Genes with similar temporal expression formed larger clusters, while those with unique kinetic patterns grouped independently.

**ORF-counting and statistics in the CHX-treated dataset.** To obtain gene expression counts, we used an overlap-based approach leveraging our in-house developed function feature.OV.from.polyC.TR(), which utilizes the tidygenomics [40] package for efficient genomic interval operations. This function identifies the most upstream overlapping genomic feature (typically CDS regions) for each exon and assigns counts to transcripts. Overlaps were determined using a genomic join strategy, where transcript features were matched to the reference annotation based on chromosomal coordinates (seqnames, start, end). The overlap criteria were set as type='any', maxgap=-1, and minoverlap=10, meaning that every overlap between a transcript and an ORF with at least 10 nucleotides in length and on the matching strand was considered a hit for the respective ORF. If multiple hits were found for a single

transcript, the most upstream hit was retained. Since all genes were sequenced under identical conditions, normalization was not required for within-sample comparisons. Statistical analyses, including a t-test and z-score analysis, were performed directly on raw counts to assess expression differences.

**Multiple sequence alignment of select genes.** The CTO and NOIR genes of equid alphaherpesvirus 1 (Genome: KJ717942.1) and pseudorabies virus (PRV; Genome: NC_001491.2) were aligned using the VectorBuilder online tool. (https://en.vectorbuilder.com/tool/sequence-alignment.html).

The R codes used to perform the analysis and generate the plots are available at: https://github.com/Balays/EHV-1-dynamic

## Results

### Dynamic EHV-1 transcriptome: general considerations

In this study, we performed a comprehensive time-course transcriptomic analysis of EHV-1, utilizing dcDNA-Seq on the ONT MinION platform. The long-read sequencing of virus-infected RK-13 cells generated a total of 3,327,022 viral reads (S1 Table). Additionally, this dataset was complemented with CAGE sequencing performed on the Illumina MiSeq platform. Our previous work [28] employed native dRNA-Seq with rigorous filtering criteria, ensuring high confidence results but excluding many detected transcripts that failed to meet the strict validation requirements. One key limitation of dRNA-Seq was its reduced accuracy in identifying transcription start sites (TSSs), partly due to the incomplete 5'-ends of the sequencing reads. To overcome these limitations, we integrated CAGE-Seq data with dcDNA-Seq reads. CAGE-Seq provides high-resolution TSS mapping, enabling more accurate identification and validation of 5'-ends. By aligning dcDNA-Seq reads that carried correct 5'-adapters and intact poly(A) tails with CAGE-derived TSS clusters, we confirmed the authenticity of previously excluded transcripts and added 169 novel ones.

For the kinetic analyses, we focused on canonical transcripts - defined as the most abundant isoforms for each viral gene - and their TSSs and TESs. Using these canonical references, we analyzed the temporal dynamics of gene expression across multiple time points, clustered genes into *de novo* kinetic classes based on their expression curves, and compared these classes with the traditional IE/E/L framework.

Finally, beyond annotating full-length transcripts and refining exact TSS/TES positions, we explored isoform switching and transcriptional overlaps, indicating how the virus orchestrates complex regulatory patterns over the infection cycle.

### The ORF64 is the only EHV-1 immediate-early gene

It has been previously established that ORF64 is the only immediate-early (IE) gene of EHV-1 [40]. However, our earlier study [28] and the current research have identified novel EHV-1 transcripts with previously unknown kinetic properties. To investigate whether the expression of these transcripts requires newly synthesized viral proteins, we treated RK-13 cells (ECACC: 00021715) with cycloheximide (CHX), a protein synthesis inhibitor, prior to infection with EHV-1. CHX were administered at concentrations of 20 and 100 μg/ml and collected samples at 6 and 8 hpi. Subsequently, we performed long-read dcDNA sequencing.

Statistical analysis confirmed that ORF64 is the sole IE gene in EHV-1 (S3 Table). A one-sample t-test demonstrated that ORF64's expression was significantly higher than that of all other viral genes under CHX treatment ($p < 10^{-47}$). Additionally, a Z-score analysis revealed that ORF64's expression was 2.84 standard deviations above the mean, identifying it as an extreme outlier. Furthermore, ORF64 exhibited 12.1-fold higher expression than the second most abundant gene (ORF67), reinforcing its unique status as an IE gene.

In PRV, the closest relative of EHV-1 with an annotated transcriptome, the homologous gene (*i.e.,180*) is also the only IE gene [41]. In contrast, other annotated alphaherpesviruses, including those in the Simplexvirus and Varicellovirus genera, have multiple IE genes among their members.

## Reannotation of the EHV-1 transcriptome

In this part of our study, we employed CAGE-Seq on an Illumina MiSeq platform to achieve high-resolution detection of the TSSs of EHV-1 transcripts. First, we cross-validated transcripts previously annotated in our laboratory using dRNA-Seq [28] with newly obtained CAGE-Seq data (S4 Table). Among the examined transcripts, 220 received the highest level of support (***) - indicating robust validation - while 34 had medium support (**) and 36 showed the lowest level of support (*) (see Methods for details). Next, we analyzed dcDNA-Seq reads that could not be assigned to previously annotated transcripts. By aligning the 5′-ends of these reads - validated by 5′-adapter sequences (using the LoRTIA pipeline) - to TSS clusters detected by CAGE-Seq (via CAGEfightR), we refined TSS annotations and identified distinct TSS peaks within broad clusters (>200 bp). The transcripts were assembled by combining these refined TSSs with TESs from our previous annotations. Transcripts were considered authentic if they had a minimum of five dcDNA-Seq reads with 5'-ends aligning to CAGE-Seq-validated TSSs. In this work, we identified 169 novel transcripts, categorized as follows: 26 ncRNAs; 80 isoforms of monocistronic transcripts (42 with longer and 38 with shorter UTRs compared to the canonical transcripts); and 18 isoforms of multicistronic transcripts (8 with longer and 10 with shorter UTRs than the canonical transcripts). Long and short UTR variants share the same ORF as the canonical transcript but differ in the length of their 5′-ends. Additionally, we identified 29 putative embedded genes encoding 5′-truncated mRNAs with in-frame ORFs that are 3′-coterminal with the larger canonical gene. Putative embedded genes may encode N-terminally truncated proteins. However, due to the phenomenon of cytoplasmic recapping [42], which can produce capped truncated transcripts, and the less robust 5′ support provided by nanopore-based sequencing methods, we applied an additional filtering criterion. Therefore, additional criteria were set for the acceptance of 5′-truncated isoforms as genuine transcripts: their TSS had to be confirmed by dRNA-Seq data, and their expression ratio had to be at least 5% relative to their canonical transcript (TSSs were determined by CAGE-Seq). We reevaluated our previously described transcripts and indicated in S4 Table whether they met our new validation criteria. Failure to meet these strict criteria does not necessarily imply that these transcripts are mere artefacts.

## Comparison of replication origin-associated RNAs of EHV-1 and PRV

These transcripts are mapped near the replication origins (OriS) in herpesvirus genomes. The replication origin-associated RNA (raRNA), CTO-S, located near the OriL has only been described in EHV-1 and PRV (S1A Fig). The NOIR ncRNA has been reported to map between the ORF64 and ORF65 transactivator genes near the Ori located at the repeat region of the short genomic region (OriS) of the EHV-1 [28]. NOIR family has been identified in Varicelloviruses, including PRV (S1B Fig), however their precise genomic locations vary among viruses [11]. As these are ncRNAs, their level of conservation is lower than that of the protein-coding regions. S1 Fig shows that the INDEL pattern of the field isolate (MdBio) and laboratory (Kaplan) PRV strains is very similar but differs from that of EHV-1, except for a long repeat insertion in the NOIR gene of the Kaplan strain at the 871–946 genomic position. S2 Fig and S3 show the OriS and OriL regions, along with the genes and transcripts in these regions, including their associated raRNAs that overlap with either replication origin in EHV-1 and PRV, respectively.

## Kinetic characterization of TSSs, TESs, and canonical transcripts

We then investigated the dynamics of TSSs, TESs, and canonical transcripts throughout the infection cycle, comparing these observations with the traditional IE, early (E), and late (L) kinetic classes. The timing of gene expression is as follows: IE genes are highly expressed at 1–2 hpi, E genes predominate at 2–6 hpi, and L genes are expressed after 6 hpi, with peak expression occurring at 8–12 hpi and later.

**TSS expression kinetics.** In this part of the study, we conducted a temporal expression analysis of the TSSs of viral transcripts (Fig 1, S4 Fig and S5 TableA). We obtained that the E genes, such as ORF20, ORF21, ORF30, ORF31, and ORF63 exhibited peak TSS activities as early as 2 hpi, followed by a gradual decline. Conversely, L genes, including ORF11, ORF14, ORF22, and ORF73, began to show substantial TSS activity starting from 4 hpi, reaching their maxima around 8–12 hpi. This pattern is consistent with the known function of these genes in either DNA synthesis (E genes), or in producing structural components for virion assembly and egress (L genes). Detailed temporal profiling further elucidated this dynamic landscape by

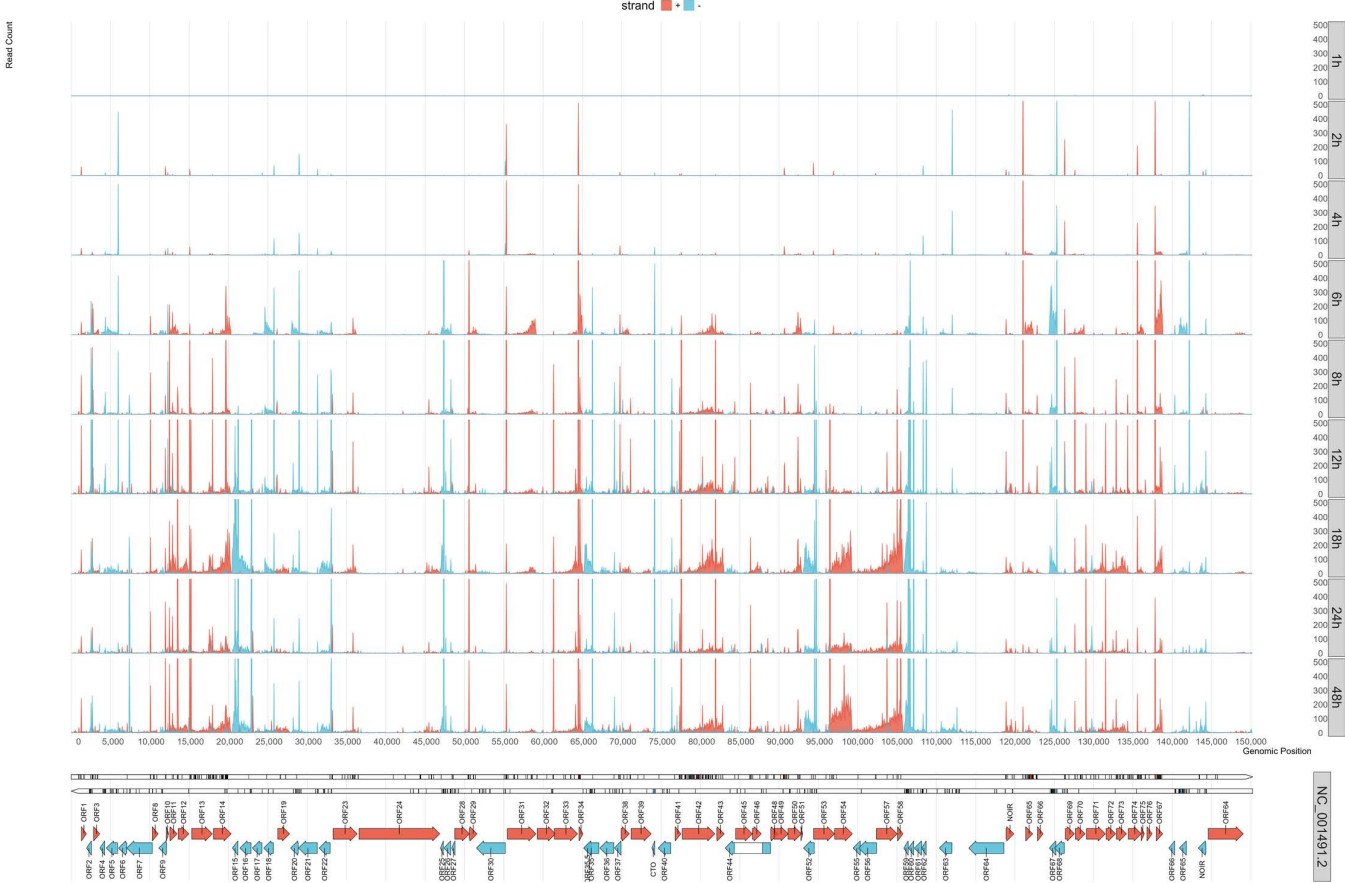

**Fig 1. Kinetics of transcription start sites of EHV-1 detected by dcDNA-Seq and validated by CAGE-Seq.** The time-course experiment utilizing dcDNA-Seq spanned 8 timepoints, ranging from 1 hour to 24 hours. TSSs were identified through LRS analysis and confirmed via CAGE-Seq. For each nucleotide, we counted the number of reads beginning at that position with their 5'-ends. We included only those reads that had clear directionality, which was determined by the presence of 5'-or 3'-adapters. Data from all three replicates were combined. We then grouped the TSS signal strength values into 50-nt segments to illustrate the distribution of TSSs. The y-axis of the graph was automatically scaled to accommodate up to 500 read counts. An image with lower-resolution details is available in S2 Fig. In the representation, genes are indicated by arrows, and the distribution of TSSs is shown in different color: red for the positive strand and blue for the negative strand. The bottom row of the image displays the CAGE-Seq counts.

pinpointing specific TSS peak times for individual transcripts. For example, ORF32 showed an early peak at 2 hpi, ORF51 at 6 hpi, and ORF19 at 8 hpi, each followed by a characteristic decline. Additional examples include ORF18, which peaked at 8 hpi, ORF28, which showed a maximum at 6 hpi and then again at 8 hpi, and ORF50, which exhibited peak activity at 4 hpi. The kinetics of the TSSs grouped by the kinetic classes are shown in S5 Fig.

To further clarify these patterns, we applied hierarchical clustering of TSS abundances throughout the infection. This approach grouped genes based on their expression trajectories rather than relying solely on predefined IE/E/L classifications. The clusters largely reinforced the TSS-based kinetics patterns: large clusters often predominantly contained E genes or L genes, reflecting the temporal shifts observed in the individual TSS analyses. For example, clusters dominated by L genes confirmed the existence of a robust late-expression phase, while clusters enriched in E genes validated an early wave of transcription closely following the IE stage. The kinetics of the TSSs grouped by *de novo* clustering are presented in S6 Fig.

However, the clustering revealed that some genes did not align neatly with the expected phases. At the single-gene level, several traditionally L genes exhibited earlier-than-anticipated TSS peaks, while some E genes maintained or regained expression at later time points. For instance, ORF38, typically classified as an L gene, displayed a TSS peak at 6 hpi - more characteristic of E kinetics -while ORF45, also considered an L gene, peaked at 12 hpi and again at 48 hpi. Similarly, ORF54, an E gene, showed a peak at 24 hpi, well beyond the window typically associated with early functions. These "misaligned" genes were grouped into heterogeneous clusters containing both E and L markers, suggesting they may represent transitional or intermediate regulatory states rather than strictly defined classes, at least according to this method of characterization. The clustering also highlighted small groups or outliers - single- or double-gene clusters - with unique dynamics patterns that do not align with the canonical IE/E/L framework. These outliers suggest that some genes may follow specialized regulatory circuits, contributing to the intricate temporal orchestration of viral gene expression. Our analysis identified 12 distinct clusters, reflecting the temporal and functional profiles of viral gene expression. Cluster_12 consists of ORF64, encoding the transcriptional regulator ICP4, underscoring its pivotal role in initiating viral transcription and, interestingly, ORF75 (US8A), traditionally considered late (likely due to detection in one replicate at 1 hpi). Early-dominant TSS clusters, such as Cluster_1 and Cluster_3, contain genes like ORF20, ORF21, ORF30, ORF53, and ORF63, involved in nucleotide metabolism and genome replication, peaking early post-infection. Intermediate clusters featuring ORF19, ORF37, ORF55, and ORF76 bridge E and L phases, indicating overlapping or transitional expression profiles. Late-dominant clusters – most notably Cluster_6 and Cluster_7 – include genes (e.g., ORF22, ORF24, and ORF42 in Cluster_6) and (e.g., ORF12, ORF13, and ORF48 in Cluster_7) that encode proteins involved in virion assembly and packaging, peaking at 8–12 hpi. Overall, these patterns reveal a continuous and overlapping temporal landscape rather than strictly partitioned IE/E/L classes.

**TES expression kinetics.** The examination of TES dynamics (Fig 2, S7 Fig and S5B Table) illustrates a complex and overlapping regulatory landscape, much like what we observed at the TSSs. Many genes align with their expected kinetic classes: E genes such as ORF20, ORF21, ORF30, ORF31, and ORF63 display TES peaks within the first few hpi, while L genes including ORF11, ORF14, ORF22, and ORF73 reach their maxima between 8 and 12 hpi. This overall pattern is consistent with the established roles of E genes in DNA replication and L genes in virion assembly. However, both the initial analysis of individual TES kinetics and the subsequent clustering based on TES usage reveal exceptions and overlapping dynamics that challenge the straightforward IE/E/L model. For example, ORF32 and ORF51, traditionally classified as L genes, exhibited earlier-than-expected TES peaks, while ORF19, categorized as

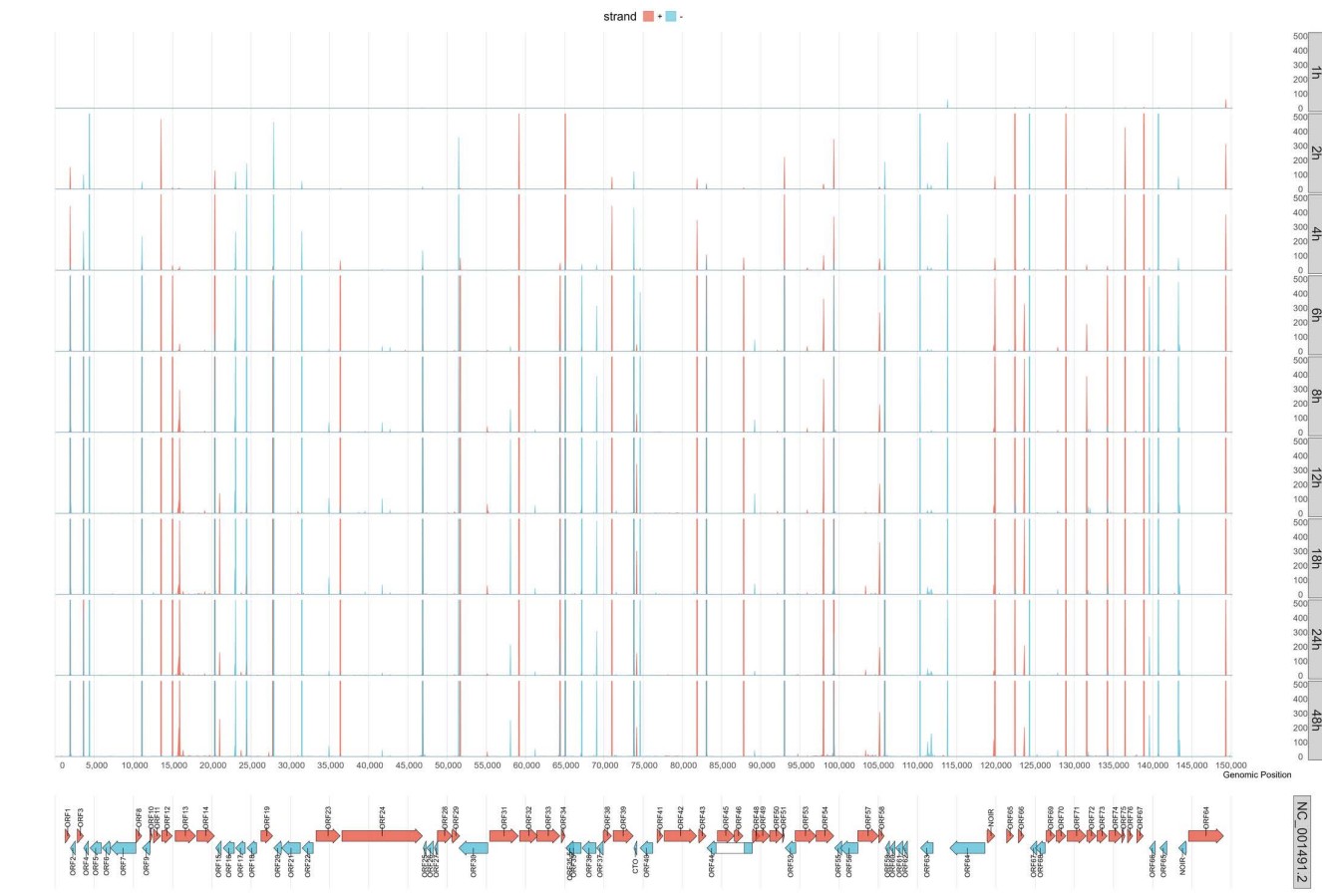

**Fig 2. Kinetics of transcription end sites of EHV-1 detected by dcDNA-Seq and validated by dRNA-Seq.** The time-course study covered 8 intervals, from 1 to 24 hours. The TESs were detected using oligo(dT) priming-based dcDNA sequencing, which subsequently were confirmed with dRNA-Seq. We counted the reads initiating from each nucleotide position with their 3'-ends, focusing only on those with clear directional cues identified by the presence of 5'- or 3'-adapters. Data from all three dcDNA-Seq replicates were merged. The aggregated read counts were then summed into 50-nt blocks to illustrate the TES distributions. The graph's y-axis was set to automatically adjust, supporting up to 500 read counts. An image with a lower-resolution view is available in S5 Fig. The diagram marks genes with arrows and color-codes the TSS distribution, using red for the positive strand and blue for the negative strand.

an E gene, showed a delayed TES maximum more characteristic of L genes. These anomalies suggest that the timing of transcript termination does not always correspond to the canonical temporal classes. However, a key reason for this anomaly is that tandem gene clusters produce co-terminal transcripts with distinct temporal expression profiles, which blur the IE/E/L boundaries in this type of analysis.

The clustering of TES expression profiles (Fig S8 and S9) revealed distinct groups of genes with shared termination dynamics. For example, clusters predominantly composed of late-expressed structural and assembly genes - such as ORF12, ORF13, and ORF14 in Cluster_6 or ORF15–18 in Cluster_8 - highlight the coordinated L-phase expression of genes encoding capsid, tegument, and packaging proteins. In contrast, clusters enriched in E genes - such as ORF7, ORF30, and ORF63 in Cluster_5 - peak during the initial stages of infection, consistent with their roles in replication and regulation. Mixed-phase clusters, such as ORF32–34 in Cluster_7 or ORF48–51 in Cluster_3, combine genes from both E and L classes, indicating that co-termination creates overlapping kinetic patterns and highlights a continuous temporal landscape rather than strictly segmented IE/E/L phases. Smaller

clusters, such as Cluster_11, containing ORF64, underscore its unique regulatory role at the TES level. Other clusters, such as those containing envelope glycoproteins and tegument proteins in Cluster_10 or multi-gene E/L sets in Cluster_1 and Cluster_12, demonstrate that transcripts with differing temporal classes can terminate together, further increasing transcriptional complexity.

**Matching TSSs and TESs.** For most herpesvirus RNAs, identifying transcript ends alone is insufficient for transcript annotation due to the co-terminal organization of tandem viral genes and the presence of alternative splicing. To investigate the linkage between TSS and TES kinetics, we analyzed lrRNA-Seq data in detail. Matching TSSs to TESs on individual transcripts allowed us to assess whether the observed differences in kinetics arose from alternative transcript isoforms, multicistronic transcripts, or other factors. Fig 3 shows each gene's abundance during the course of the infection, as assessed by the viral-read-normalized canonical transcript counts, according to their kinetic classes. For genes where TSS and TES dynamics differed, our analysis revealed that the discrepancies could often be attributed to the complex transcriptional landscape of EHV-1. The virus produces a variety of transcript isoforms, including alternative TSSs and sometimes TESs, as well as multicistronic and overlapping transcripts (Fig 4, S10 Fig and S5C Table). For example, both ORF38 and ORF50, expected to show late kinetics, exhibited early peaks, although their TSS peaked at 4 and 8 hpi, while their TES at 6 hpi. This misalignment is in the case of ORF38 the result of the elevated expression of other, mainly complex transcripts (overlapping ORF35–37 from the other strand) that terminate at the same TES, such as ORF37-ORF38-CX-Long-2 (Fig 4A) and possible transcriptional noise from other non-validated TSSs in this region. In the case of ORF50 (S10B Fig), this could be attributed to a more complex differential transcript expression pattern consisting of mainly ORF50-ORF51-Canonic and ORF50-ORF51-PC-Long-2. Conversely, ORF67, which showed an early TSS peak, had TES dynamics more consistent with its L gene classification (S10C Fig). While its presence at 1 hpi could be attributed to transcriptional noise affecting this very early time point, at 2 hpi it is more likely due to the highly efficient early activation of its promoter. The discrepancy between its TSS and TES kinetics can be explained by the large variety of potential mRNAs identified in this region, each carrying 5′-truncated ORFs, albeit with individually low counts. Our detailed mapping confirms that the discrepancies between TSS and TES kinetics are primarily due to the production of multiple transcript isoforms and the complex arrangement of transcription units in the EHV-1 genome. This underscores the importance of examining full-length transcript structures when interpreting gene expression dynamics, a task for which LRS techniques are the only reliable approach.

**Gene-level clustering of canonical transcripts.** Clustering canonical full-length transcripts provides a cleaner view of EHV-1's transcriptional program. We categorized the viral genes into three *de novo* kinetic clusters (**Fig 5**).

**Heterogeneous clusters.** Cluster_1 contains E genes (e.g., ORF20, ORF21, ORF31, ORF61), L genes (e.g., ORF9, ORF38, ORF50), and genes with undetermined kinetics. It peaks around 2–4 hpi, indicating "leaky-late" activity within an E-expression context. Similarly, Cluster_3 is E-biased (e.g., ORF5, ORF7, ORF30, ORF53, ORF63) but includes a few L genes (e.g., ORF10, ORF17). This pattern reinforces the idea that some L transcripts are detectable at low levels early on, blending replication and assembly factors in a transitional manner between 2–4 hpi, possibly as a result of stochastic transcriptional activity.

**Robust late-dominant clusters.** Cluster_2 predominantly comprises L genes (e.g., ORF11, ORF14, ORF18, ORF26, ORF28, ORF29, ORF3, ORF39, ORF40, ORF68, ORF73, ORF76) along with a few genes of undetermined kinetics (e.g., ORF2, ORF75). This cluster aligns with a robust L-phase expression wave emerging after 6–8 hpi. Similarly, Cluster_5 consists

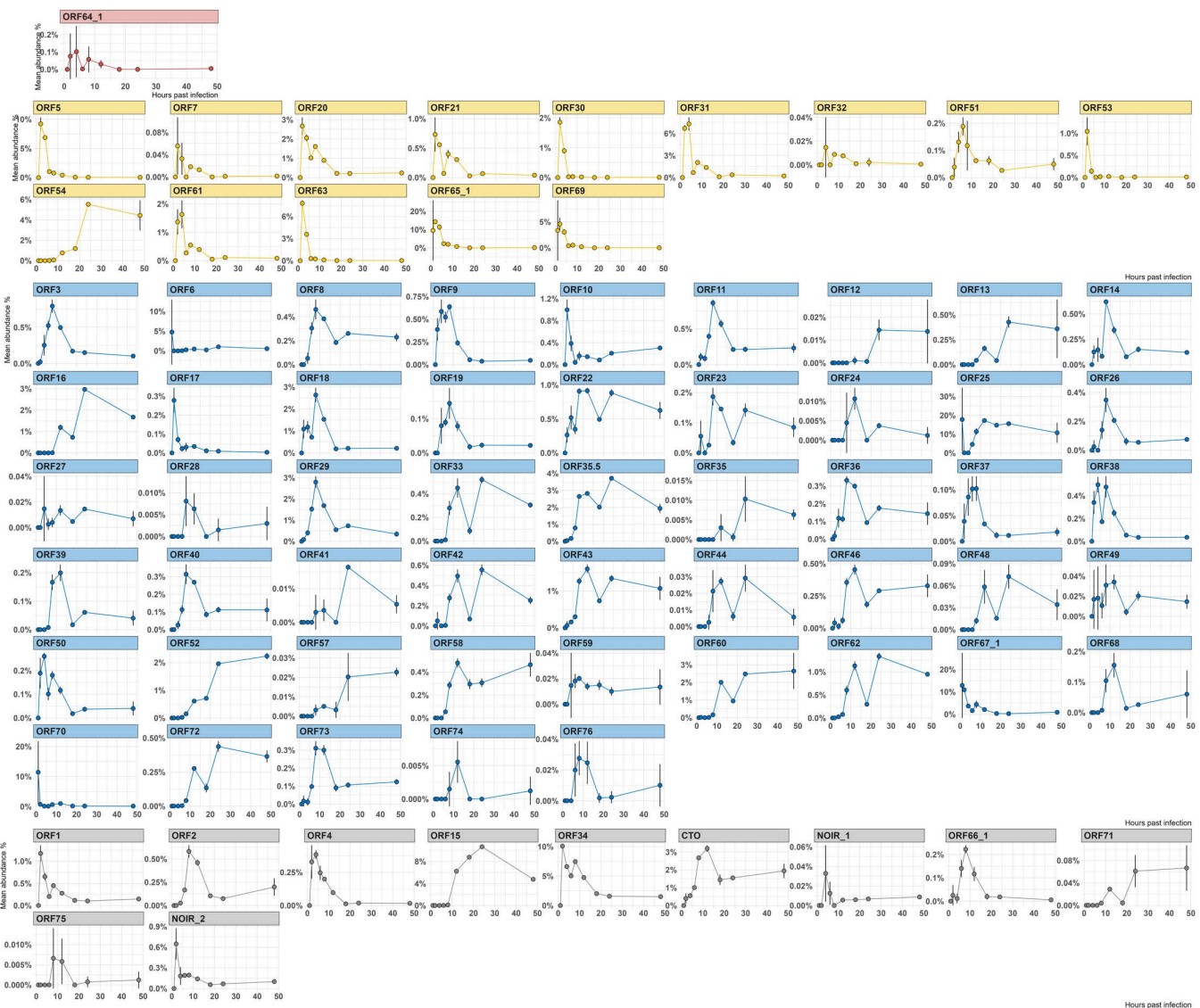

**Fig 3. Kinetic profiling of canonical EHV-1 transcripts using total viral read counts for normalization according to kinetic classes.** This figure illustrates the kinetic profiling of canonical EHV-1 transcripts, utilizing the total viral read counts per sample for normalization. The analysis included only those reads that aligned with both the canonical TSS of genes at their 5'-ends and the canonical TES of genes at their 3'-ends (allowing a deviation of +/- 10 nucleotide for both alignments). This method aggregated the counts of canonical transcripts for each gene in every sample. The mean values are represented as points, and standard deviations (SD) as lines, plotted on the y-axis as the ratio of transcript abundance for each gene. The x-axis represents time post-infection (hours). The panels are color-coded based on kinetic transcription phases: blue for immediate early (IE), orange for early (E), green for late (L), and red for unknown phases. This provides a visual distinction among different transcriptional dynamics throughout the infection.

of late structural and packaging components (e.g., ORF22, ORF23, ORF25, ORF33, ORF35.5, ORF36, ORF42, ORF43, ORF44, ORF46, ORF58, ORF62), which steadily produce virion-related proteins during mid-to-late infection. Clusters_6 and _7 are also late-dominated but include notable exceptions. For instance, Cluster_6 primarily consists of L genes (e.g., ORF12, ORF13, ORF16, ORF35, ORF41, ORF48, ORF52, ORF57, ORF60, ORF72) involved in tegument formation and packaging. However, it also includes one E gene (ORF54) and one of undetermined kinetics (ORF71), reflecting the complexity of L-phase expression. Similarly, Cluster_7 combines predominantly L genes with overlapping temporal profiles.

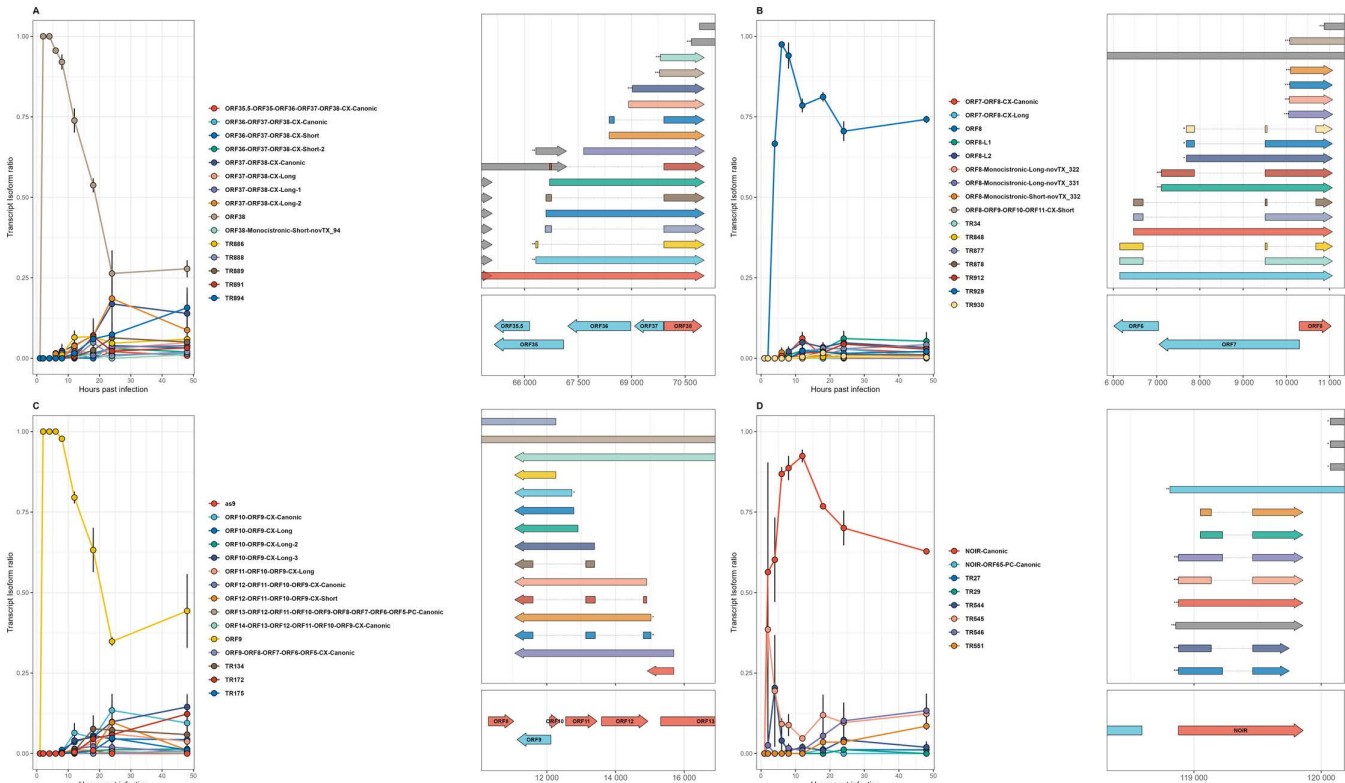

**Fig 4. Dynamics of transcript isoform usage in splice-containing EHV-1 genes over the course of infection.** This figure presents the splicing dynamics within EHV-1 for the genes (A) ORF38, (B) ORF8, (C) ORF9, (D) NOIR, (E) ORF58 and (F) ORF53. The right side of each panel shows the transcript annotations, along with their parent genes and genomic locations displayed below them, with light red indicating positive-strand genes and light blue indicating negative-strand genes. The analysis focused on transcripts that matched exactly, allowing a deviation of +/- 2 nucleotides (nt) for splice junctions and +/- 10 nucleotides for the start and end positions of transcripts. Asterisks indicate the CAGE-Seq significance level for each reference transcript. On the left side of each plot, the temporal trends of these transcript isoforms are depicted, with averages and standard deviations (SD) calculated for each time point post-infection, based on read count data from the dcDNA-Seq. Each data point is linked by lines to demonstrate the progression over time. The transcript isoforms, are color-coded according to their distinct isoforms, with these color matching those used for the points and lines in the left panel. The isoform counts were normalized against the total number of isoform counts for each gene in each sample to calculate the ratio of each isoform. Isoforms on the right side are colored grey, if they not originate from the given gene and thus were not included in the isoform ratio calculation.

**Special clusters.** Cluster_4 and Cluster_12 contain fewer genes and exhibit diverse expression kinetics. Cluster_4 includes both L (e.g., ORF6, ORF67) and E genes (e.g., ORF65), indicating subtle overlaps even in smaller sets. Meanwhile, Clusters_8, _9, and _10 refine subsets of late genes or highlight unique outliers. Notably, Cluster_10 contains the sole IE ORF64 gene. Although no canonical full-length transcripts were detected at 1 hpi - likely due to technical challenges capturing this very long RNA - the clear isolation of ORF64 within its own cluster underscores its distinct temporal regulation (S10D Fig).

Collectively, these clusters confirm that while the IE/E/L scheme provides a broad framework, actual gene expression patterns form a continuous and overlapping temporal gradient. S11 Fig shows the *de novo* clusters of EHV-1 genes, along with their kinetic classes. By integrating these analyses, we observe a consistent narrative: while many EHV-1 genes follow the classical IE, E, and L progression, a subset displays more complex or hybrid kinetics. Gene-level TSS analysis highlights individual timing anomalies, while clustering places these anomalies in a broader context, revealing that they form part of overlapping transcriptional waves rather than discrete phases. These findings suggest that EHV-1 gene regulation is

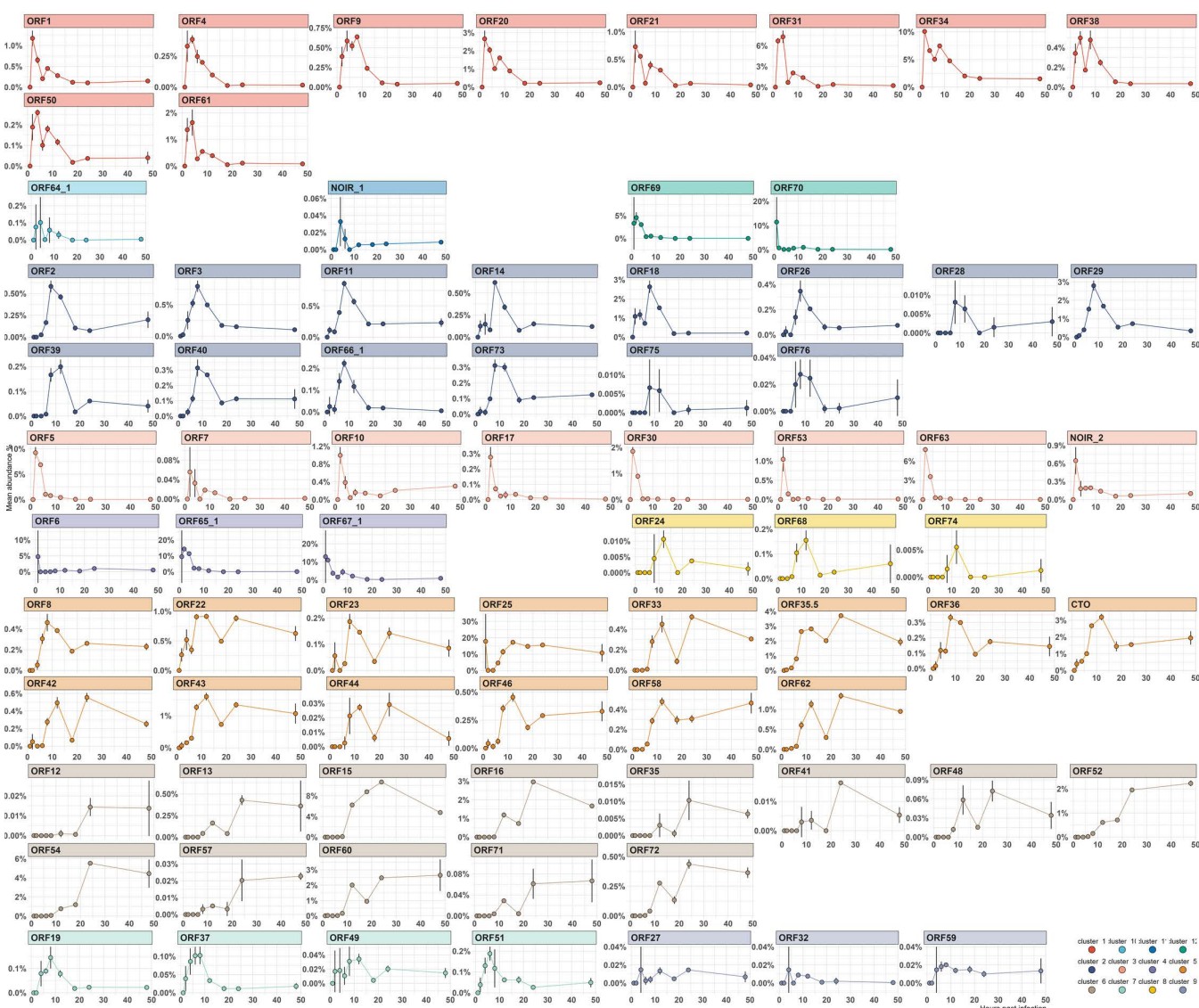

**Fig 5. Kinetic profiling of canonical EHV-1 transcripts using total viral read counts for normalization according to *de novo* kinetic clusters.** This figure illustrates the kinetic profiling of canonical EHV-1 transcripts, utilizing the total viral read counts per sample for normalization. The analysis included only those reads that aligned with both the canonical TSS of genes at their 5'-ends and the canonical TES of genes at their 3'-ends (allowing a deviation of +/- 10 nucleotides for both alignments). This method aggregated the counts of canonical transcripts for each gene in every sample. The mean values are represented as points, and standard deviations (SD) as lines, plotted on the y-axis as the ratio of transcript abundance for each gene. The x-axis represents time post-infection (hours). Each cluster is colored according to its *de novo* kinetic cluster membership. The color-coding for the clustering is shown in the bottom right panel. This figure provides a visual distinction among different transcriptional dynamics, according to the gene's relative abundance throughout the infection.

multifaceted, with certain genes bridging temporal classes and potentially serving specialized regulatory or structural roles at unconventional times during infection.

## Dynamics of spliced transcript expression

The splice sites of EHV-1 transcripts were previously identified in our laboratory using native RNA sequencing. We detected splice sites in the following genes: ORF8, ORF9, ORF38, ORF47–44, ORF53, ORF54, ORF58, ORF65, and within the NOIR family of non-coding

transcripts. Fig 4 and S12 Fig illustrate these genes. For several of them, we observed notable shifts in the ratio of spliced to non-spliced transcripts over the course of the infection.

In the gene ORF9, the combined ratio of spliced transcripts TR134 and TR172 - both sharing an intron and carrying a 5′-truncated ORF with in-frame ATG - remained at 0% (mean = 0.0) from 1–8 hpi (**Fig 4C**). Their ratio began to rise at 12 hpi (mean = 1.03%, SD = 0.84%), increased significantly at 18 hpi (mean = 11.49%, SD = 5.65%), continued to increase at 24 hpi (mean = 12.85%, SD = 3.60%), and peaked at 48 hpi (mean = 17.10%, SD = 3.64%). Other non-spliced isoforms also elevated compared to the canonical ORF9 transcript, which dropped from 100% early at 2–4 hpi with 100% expression (mean = 100.0), to only 41.69% (SD = 10.30%) by 48 hpi.

In the case of ORF38 (**Fig 4A**), we saw a very similar pattern, albeit the canonical transcript decreased in proportion even more, to 26.36% at 24 hpi and 27.70% at 48 hpi, reflecting a sharper decline compared to ORF9. This decrease in the canonical transcript was accompanied by a marked increase in the ratios of spliced transcripts TR886, TR888, TR889, and TR891, whose combined ratios rose significantly at 24 hpi and remained elevated at 48 hpi. These spliced transcripts share a second exon (which carries the ORF) and an identical intron but differ in their first exons, which define distinct 5′-UTRs. This suggests that the splicing process is tightly regulated, driving transcript diversity and contributing to the sharp decline of the canonical transcript.

The spliced transcripts TR148, TR150, TR152, TR154, and TR3 of ORF65 showed a common expression pattern. Their combined ratio peaked early at 2 hpi with 66.59%, remained high at 4 hpi (66.37%) and 8 hpi (59.76%), declined to 40.63% by 6 hpi, and decreased further through 12 hpi (59.05%), 18 hpi (15.97%), 24 hpi (17.22%), and 48 hpi (20.01%), reflecting their predominant expression during the early stages of infection (S12A Fig). These spliced transcripts share the second exon, which carries the ORF, but differ in their intron donor sites or the 5′-ends of their first exons.

In ORF44 (S12B Fig), the canonical and short non-spliced isoforms exhibited transient peaks at 8 hpi (8.28% and 3.03%, respectively) before declining to negligible levels by 48 hpi. Spliced isoforms displayed greater diversity and temporal variation, highlighting the complexity of EHV-1 splicing. TR418 peaked early at 6 hpi (77.78%), whereas TR419, supported by CAGE, reached its peak later at 12 hpi (25.74%) and maintained moderate expression levels through 48 hpi (6.60%). TR416, which shares the same second exon and intron but has a short first exon, also peaked at 12 hpi (11.26%). Notably, TR421 (extending into ORF49) peaked at 4 hpi (33.33%), while TR423 and TR424 (extending into the antisense strand of ORF50) peaked later, reaching 4.48% and 11.03% at 48 hpi, respectively. These splicing patterns align with our earlier findings [28], which demonstrated that EHV-1 undergoes a higher frequency of splicing events compared to related alphaherpesviruses, particularly in unique genomic regions such as ORF44. Furthermore, the splicing in ORF44 extends into adjacent genomic areas, representing a distinct characteristic of this virus.

NOIR, a novel non-coding raRNA has recently been described [28]. The canonical NOIR transcripts (NOIR-canonical and NOIR-ORF65-PC-canonical) showed higher expression at all time points, peaking at 12 hpi (46.08% vs. 3.79% for isoforms), while the splice isoforms contribute modestly throughout but increase relatively at later stages, reaching 14.98% at 24 hpi and 18.46% at 48 hpi compared to 35.02% and 31.15% for the canonical transcripts (Fig 4D). Similar to other genes, the spliced NOIR transcripts share a second exon and an identical intron acceptor site but differ in their intron donor sites and/or TSS. In ORF58 (S12D Fig), the canonical ORF58 transcript peaked at 6 hpi (96.30%) and declined steadily thereafter. ORF58-L1 exhibited an early peak at 2 hpi (33.33%) before decreasing to negligible levels. The spliced transcript TR1072 showed negligible expression early on but increased significantly at

24 hpi (16.54%) and remained prominent at 48 hpi (15.12%), highlighting distinct temporal expression patterns within this gene.

## Dynamics of transcriptional isoform switching in selected genes

By examining changes in isoform dominance over time, we uncovered dynamic patterns that highlight how the virus controls gene expression at various stages of infection. Fig 6 illustrates these transcripts and their shifting proportions, showing that even canonical transcripts can be temporarily superseded by truncated or alternatively terminated isoforms. For example, the expression dynamics of ORF19 revealed a transition from an early dominance of the canonical transcript, which peaked at 1 hpi (100%) and declined to 14.59% by 48 hpi, to increasing contributions from the combined complex isoforms (CX). These isoforms, which overlap completely with the coding sequence of ORF18 on the opposite strand, rose from negligible levels at 1–4 hpi to 41.44% at 12 hpi, peaking at 72.21% by 48 hpi (Fig 6A). Another example is the canonical ORF40 transcript, which dominated expression during the early stages, peaking at 6 hpi (97.78%) before declining sharply. Alternatively terminated (AT) isoforms contributed minimally during the early stages but increased in abundance later, with AT2 peaking at 37.42% at 24 hpi, while other AT isoforms remained relatively low in expression throughout the time course (Fig 6B). The canonical transcript of ORF13 (ORF13-ORF14-PC-Canonic) showed increasing expression from 2 hpi (8.33%) to a peak at 48 hpi (73.45%), while

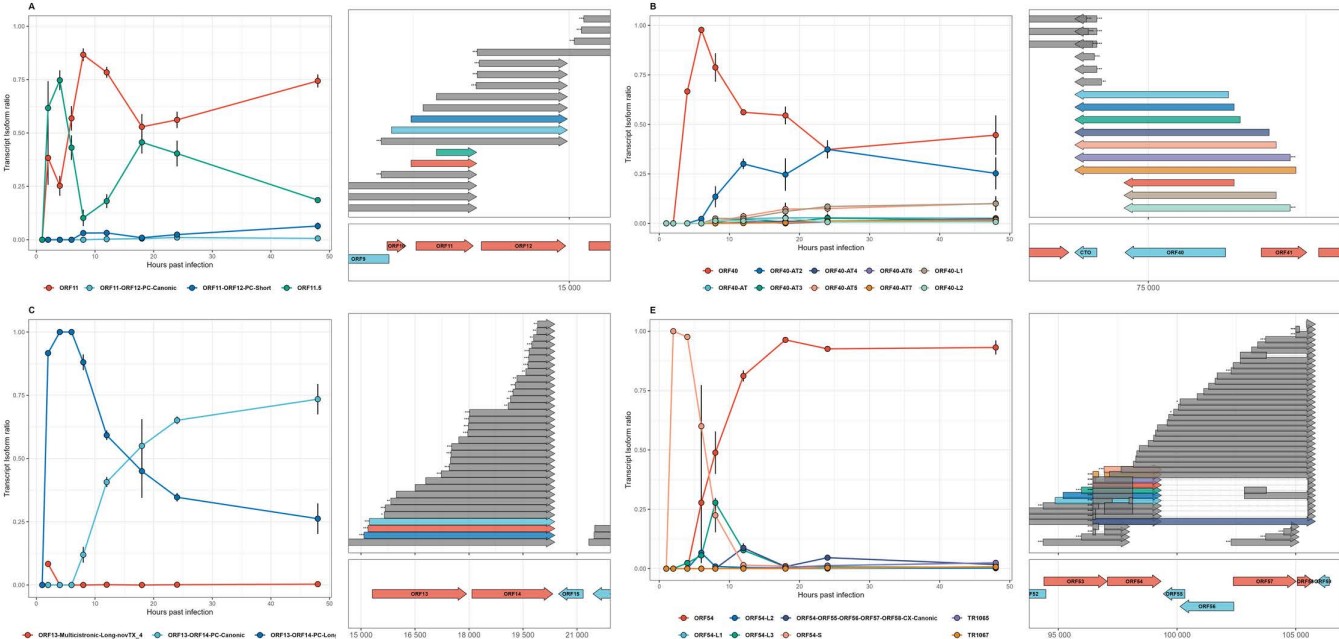

**Fig 6. Kinetics of transcript variants for isoform switching genes.** This figure illustrates the dynamics of different transcript isoforms for the selected EHV-1 genes (A) ORF11, (B) ORF19, (C) ORF13, (D) ORF40, (E) ORF14, and (F) ORF54. The right side of each panel shows the transcript annotations, along with their parent genes and genomic locations displayed below them, with light red indicating positive-strand genes and light blue indicating negative-strand genes. The analysis focused on transcripts that matched exactly, allowing a deviation of +/- 2 nucleotides (nt) for splice junctions and +/- 10 nucleotides for the start and end positions of transcripts. Asterisks indicate the CAGE-Seq significance level for each reference transcript. On the left side of each plot, the temporal trends of these transcript isoforms are depicted, with averages and standard deviations (SD) calculated for each time point post-infection, based on read count data from the dcDNA-Seq. Each data point is linked by lines to demonstrate the progression over time. The transcript isoforms, are color-coded according to their distinct isoforms, with these color matching those used for the points and lines in the left panel. The isoform counts were normalized against the total number of isoform counts for each gene in each sample to calculate the ratio of each isoform. Isoforms on the right side are colored grey, if they not originate from the given gene and thus were not included in the isoform ratio calculation.

the long isoform (ORF13-ORF14-PC-Long) dominated early expression, peaking at 4 hpi (100%) and gradually declining to 26.22% at 48 hpi (**Fig 6C**). In ORF14, transcript dynamics revealed distinct patterns and isoform switching among the canonical isoform, the long variants (L), and the truncated variant (ORF14.5, representing a 5′-truncated transcript). The canonical isoform dominated early expression, peaking at 63.89% at 2 hpi, and did not regain prominence later, showing a steady decline. The long variants (L) contributed modestly, with specific peaks such as L4 at 18 hpi (3.89%), while others like L1 and L3 remained low throughout, peaking below 1% by 48 hpi. The truncated variant ORF14.5 dominated the mid-phase, peaking at 78.54% at 6 hpi and remaining the most abundant transcript until 18 hpi, before declining to 24.44% at 48 hpi (S12E Fig). In the gene ORF11, the canonical ORF11 transcript peaked at 8 hpi (86.65%) before declining, while its 5′-truncated ORF-carrying ORF11.5 isoform showed an inverse pattern, peaking early at 4 hpi (74.72%) and decreasing steadily thereafter (S12F Fig). For ORF54, the short (S) isoform dominated early expression, peaking at 2 hpi (100%) and declining sharply after 6 hpi, while the canonical transcript emerged later, surpassing the short isoform at 8 hpi (48.85% vs. 22.47%) and dominating mid-to-late stages, peaking at 18 hpi (96.36%) and maintaining high levels through 48 hpi (93.17%) (S12C Fig).

### Dynamics of transcriptional overlaps

We used the raw sequencing reads for demonstrating the extreme complexity of transcriptional overlaps in EHV-1. S13 Fig shows that the extent of these overlaps increases as the infection progresses. The co-expression of adjacent genes inevitably results in conflicts between transcription machineries at overlapping regions during transcription. This potential genome-wide regulatory mechanism is very difficult to study with the current technology.

## Discussion

The last couple of years have witnessed significant advancements in sequencing technologies [43]. Full-length, lrRNA-Seq methods have revolutionized transcriptome research, particularly in organisms with small genomes. This has revealed that viral transcriptomic structures are far more complex than previously thought [11]. Discoveries include a wide array of RNA molecules, such as extended 5′-UTR isoforms, polygenic and complex transcripts (containing at least one gene on an opposite orientation), 5′-truncated mRNAs containing in-frame ORFs, and read-through transcripts [21,44–46]. Recent studies have demonstrated that the occurrence of nested genes within a larger canonical gene is more prevalent in viruses than previously believed [23,46,47].

RNA degradation, library preparation, and sequencing itself can cause 5'-truncation in transcripts, potentially leading to the misidentification of TSSs [48]. To address this issue, we utilized CAGE sequencing via the Illumina MiSeq platform, a standard approach for identifying the 5'-ends of capped RNA molecules. Although CAGE-Seq is generally reliable, it has the potential for detecting some fraction of truncated RNA molecules. This possibility arises because, in mammalian cytoplasm degraded RNAs can be capped by special host enzymes [49]. Although these truncated transcripts originate from biological processes and may even be functional, we attempted to minimize this form of 'noise' by setting a high score threshold for recognizing them as true TSSs resulting from transcription.

Polycistronism is a common feature in bacteria and viruses, but it is rare in eukaryotes. In prokaryotes and bacteriophages, a ribosomal binding site on the mRNAs, known as the Shine-Dalgarno sequence, facilitates the translation of downstream genes in polycistronic RNA molecules. Many small-genome eukaryotic viruses have evolved various mechanisms, such as internal ribosome entry sites, ribosomal frameshifting, or leaky ribosomal scanning

[50]. In herpesviruses, co-oriented genes often form clusters that produce transcripts with shared downstream sequences and unique 5′-exons, following a pattern like 'abcd', 'bcd', 'cd', and 'd', where 'a' is the most upstream gene and 'd' is the most downstream. The role of polygenic transcripts in large DNA viruses remains unclear, as translation from downstream genes has been rarely documented [51,52].

We have previously reported that EHV-1 exhibits more frequent splicing events compared to related alphaherpesviruses [28]. Transcripts of ORF44 (homolog of HSV *ul15* gene), ORF65 (homolog of HSV *us1* gene), and NOIR are spliced in other alphaherpesviruses as well. However, EHV-1 uniquely features splicing in different genomic regions, such as ORF6/12, ORF35/39, and ORF53/58. Additionally, the splicing observed in ORF44 extends to adjacent genomic areas, including ORF49/50, which is a distinctive characteristic of this virus.

An ongoing challenge in long-read RNA sequencing studies is that most existing pipelines prioritize the most abundant isoforms and rely on reference annotations primarily designed for eukaryotic transcriptomes, often overlooking the complexities typical of viral genomes. Recent benchmarks, for instance, have focused on synthetic or mammalian datasets without addressing the extensive transcriptional overlap, alternative TSSs and complex transcripts that frequently occur in viruses [53]. Tools such as StringTie2 and gff-compare, while effective for annotating standard eukaryotic transcripts, often assign shorter isoforms to the longest one. This issue is exacerbated in viruses, where multiple transcription initiation and overlapping transcripts create a dense genomic landscape. NAGATA, a pipeline reportedly applicable to viruses, has shown promise for native RNA sequencing data [38]. However, it explicitly discards 5′ soft-clipped reads, making it unsuitable for dcDNA-Seq libraries that rely on 5′-adapter sequences to orient alignments. In contrast, our LoRTIA pipeline accommodates these 5′-adapter sequences, using them to determine strand information and effectively process dcDNA-Seq data. Nevertheless, LoRTIA also faces the fundamental challenge of distinguishing genuine 5′-truncated transcripts from artifacts introduced by incomplete reverse transcription or cytoplasmic mRNA recapping. The latter can generate novel 5′-ends that may be biologically meaningful [42], but these risk being misidentified as legitimate viral TSSs without careful filtering.

In view of these constraints, our study focused on capturing canonical full-length viral transcripts while also monitoring alternative TSSs and TESs at lower abundance. By integrating multiple data sources (dcDNA-Seq, dRNA-Seq, and CAGE-Seq) and using different tools - including LoRTIA for dcDNA-Seq libraries and NAGATA for dRNA-Seq data - we applied stringent criteria, particularly for 5′-truncated transcripts, to minimize false positives without overlooking potentially functional low-abundance isoforms. Additionally, we performed a peak analysis on TSS clusters identified from the CAGE data (CAGEfightR), refining 5′-boundaries using dcDNA-Seq read counts. Together, these measures provide a robust, flexible strategy for accurately annotating viral transcripts under current methodological and computational constraints.

In this work, we integrated dcDNA-Seq and CAGE-Seq data to create a comprehensive, time-resolved map of the EHV-1 transcriptome. By leveraging the strengths of each method - CAGE-Seq for high-resolution TSS identification and dcDNA-Seq for capturing full-length transcripts - we validated previously excluded transcripts and identified a substantial number of new isoforms. We documented dynamic patterns of isoform switching, highlighting the complexity of EHV-1 transcriptome beyond previous understanding. This integrated approach also enabled us to cluster genes into *de novo* kinetic classes, revealing overlapping temporal waves of expression that go beyond the traditional IE/E/L framework.

The implications of our findings are significant for understanding the regulatory strategies of EHV-1. The presence of numerous transcript isoforms and intricate splicing dynamics

suggests that the virus employs multiple layers of transcriptional and post-transcriptional control. Temporal shifts in isoform prevalence, along with alternative TSSs and TESs, indicate that EHV-1 enables the virus to respond flexibly to host conditions, optimize resource utilization, and orchestrate the production of viral components for efficient replication and spread.

We believe that the significance of these isoforms arises not only from their coding capacity but also from their ability to physically inhibit the transcription of other genes through the process of transcriptional interference [11]. For instance, the early increase in the 5'-truncated isoform of ORF11 could indicate a regulatory mechanism to quickly produce a necessary protein without the full-length transcript, while the later increase may reflect a need for the complete protein function in later stages. On the other hand, the steady increase in ORF40 AT isoforms (TES variants) suggests a role in interfering with the transcription initiation of its adjacent gene, the CTO, which is the most abundant non-coding transcript.

Taken together, our results emphasize that EHV-1 gene regulation is governed by a sophisticated and multi-layered transcriptional program. By delineating the full complement of viral transcripts and their temporal patterns - including isoform switching – our data provide a solid foundation for future studies to unravel the molecular mechanisms underlying viral replication and pathogenesis.

## Supporting information

**S1 Figure.  Comparison of replication origin-associated transcripts.** This illustration compares the sequences of raRNAs [(A) CTO-S; (B) NOIR] from EHV-1 with those of two PRV strains (Kaplan and MdBio).
(JPG)

**S2 Figure.  raRNAs of the OriL and OriS replication origins of EHV-1.** The OriL (upper panel) and OriS (lower panel) replication origins of equid alphaherpesvirus 1 and the genes located in their surrounding regions are shown. In the figure, transcripts and genes from the forward strand are shown in red, while those from the reverse strand are shown in blue. The replication-associated RNA (raRNA) molecules overlapping the replication origin are highlighted in green, while the Ori site is highlighted in purple.
(PDF)

**S3 Figure.  raRNAs of the OriL and OriS replication origins of PRV.** The OriL (upper panel) and OriS (lower panel) replication origins of pseudorabies virus, along with the genes in their surrounding regions, are depicted. In the figure, transcripts and genes on the forward strand are displayed in red, while those on the reverse strand appear in blue. The replication-associated RNA (raRNA) molecules that overlap the replication origin are highlighted in green, while the Ori site is marked in purple.
(PDF)

**S4 Figure.  Kinetics of transcription start sites of EHV-1 detected by dcDNA-Seq and validated by CAGE-Seq.** Similar to Figure 1, this plot shows the 5'-end distribution along the viral genome for each time-point group. The mean values for each time-point group were calculated and merged into 50-nt sized bins for visualization. Each facet corresponds to a time-point group, and the y-axis scale is determined independently (scale="free_y" in ggplot).
(JPG)

**S5 Figure.  Kinetic profiling of canonical EHV-1 TSSs according to kinetic classes.** This figure illustrates the kinetic profiling of canonical EHV-1 TSSs, utilizing the total viral read counts per sample for normalization. The analysis included only those reads that aligned with the canonical TSS of genes at their 5'-ends (allowing a deviation of +/- 10 nucleotides). The

mean values are represented as points, and standard deviations (SD) as lines, plotted on the y-axis as the ratio of TSS abundance for each gene. The x-axis represents time post-infection (hours). The panels are color-coded based on kinetic transcription phases: blue for immediate early (IE), orange for early (E), green for late (L), and red for unknown phases. This provides a visual distinction among different TSS dynamics throughout the infection.
(JPG)

**S6 Figure. Kinetic profiling of canonical EHV-1 TSSs according to de novo kinetic clusters.** This figure illustrates the kinetic profiling of canonical EHV-1 TSSs, utilizing the total viral read counts per sample for normalization. The analysis included only those reads that aligned with the canonical TSS of genes at their 5'-ends (allowing a deviation of +/- 10 nucleotides). The mean values are represented as points, and standard deviations (SD) as lines, plotted on the y-axis as the ratio of TSS abundance for each gene. The x-axis represents time post-infection (hours). Each cluster is colored according to its *de novo* kinetic cluster membership. The color-coding for the clustering is shown in the bottom right panel. This figure provides a visual distinction among different transcriptional dynamics, according to the gene's relative TSS abundance throughout the infection.
(JPG)

**S7 Figure. Kinetics of transcription end sites of EHV-1 detected by dcDNA-Seq and validated by dRNA-Seq.** Similar to Figure 2, this plot shows the 5'-end distribution along the viral genome in each time-point groups. The mean values for each time-point group was calculated and merged into 50-nt sized bins for the visualization. Each facet represents a time-point group, with the y-axis scale set independently using scale="free_y" in ggplot.
(JPG)

**S8 Figure. Kinetic profiling of canonical EHV-1 TESs According to kinetic classes.** This figure illustrates the kinetic profiling of canonical EHV-1 TESs, utilizing the total viral read counts per sample for normalization. The analysis included only those reads that aligned with the canonical TES of genes at their 5'-ends (allowing a deviation of +/- 10 nucleotides). The mean values are represented as points, and standard deviations (SD) as lines, plotted on the y-axis as the ratio of TES abundance for each gene. The x-axis represents time post-infection (hours). The panels are color-coded based on kinetic transcription phases: blue for immediate early (IE), orange for early (E), green for late (L), and red for unknown phases. This provides a visual distinction among different TSS dynamics throughout the infection.
(JPG)

**S9 Figure. Kinetic profiling of canonical EHV-1 TESs according to de novo kinetic clusters.** This figure illustrates the kinetic profiling of canonical EHV-1 TESs, utilizing the total viral read counts per sample for normalization. The analysis included only those reads that aligned with the canonical TSS of genes at their 5'-ends (allowing a deviation of +/- 10 nucleotides). The mean values are represented as points, and standard deviations (SD) as lines, plotted on the y-axis as the ratio of TES abundance for each gene. The x-axis represents time post-infection (hours). Each cluster is colored according to its *de novo* kinetic cluster membership. The color-coding for the clustering is shown in the bottom right panel. This figure provides a visual distinction among different transcriptional dynamics, according to the gene's relative TES abundance throughout the infection.
(JPG)

**S10 Figure. Kinetics of transcript isoforms for selected genes.** This figure illustrates the dynamics of different transcript isoforms for the selected EHV-1 genes (A) ORF23, (B) ORF51, (C) ORF67, (D) ORF64. The right side of each panel shows the transcript annotations,

along with their parent genes and genomic locations displayed below them, with light red indicating positive-strand genes and light blue indicating negative-strand genes. The analysis focused on transcripts that matched exactly, allowing a deviation of +/- 2 nucleotides (nt) for splice junctions and +/- 10 nucleotides for the start and end positions of transcripts. Asterisks indicate the CAGE-Seq significance level for each reference transcript. On the left side of each plot, the temporal trends of these transcript isoforms are depicted, with averages and standard deviations (SD) calculated for each time point post-infection, based on read count data from the dcDNA-Seq. Each data point is linked by lines to demonstrate the progression over time. The transcript isoforms, are color-coded according to their distinct isoforms, with these color matching those used for the points and lines in the left panel. The isoform counts were normalized against the total number of isoform counts for each gene in each sample to calculate the ratio of each isoform. Isoforms on the right side are colored grey, if they not originate from the given gene and thus were not included in the isoform ratio calculation.
(JPG)

**S11 Figure.** *De novo* **clustering of EHV-1 genes.** The clustering of EHV genes is based on the normalized canonical transcript counts (reads spanning from the canonical TSS tot the TES), compared to the traditional kinetic classification. The rows show the *de novo* cluster memberships, while the color represent the kinetic classes.
(JPG)

**S12 Figure.** **This figure presents the splicing dynamics within EHV-1 for the genes (a) ORF65, (b) ORF44.** The right side of each panel shows the transcript annotations, along with their parent genes and genomic locations displayed below them, with light red indicating positive-strand genes and light blue indicating negative-strand genes. The analysis focused on transcripts that matched exactly, allowing a deviation of +/- 2 nucleotides (nt) for splice junctions and +/- 10 nucleotides for the start and end positions of transcripts. Asterisks indicate the CAGE-Seq significance level for each reference transcript. On the left side of each plot, the temporal trends of these transcript isoforms are depicted, with averages and standard deviations (SD) calculated for each time point post-infection, based on read count data from the dcDNA-Seq. Each data point is linked by lines to demonstrate the progression over time. The transcript isoforms, are color-coded according to their distinct isoforms, with these colors matching those used for the points and lines in the left panel. The isoform counts were normalized against the total number of isoform counts for each gene in each sample to calculate the ratio of each isoform. Isoforms on the right side are colored grey, if they don'toriginate from the given gene and thus were not included in the isoform ratio calculation.
(JPG)

**S13 Figure.** **Dynamics of transcriptional overlaps.** The annotation of transcripts is based on strict criteria, resulting in the exclusion of a significant number of viral reads. To circumvent this loss, we used the raw sequencing reads to illustrate the true extent of transcriptional overlaps created by genes arranged in divergent and convergent orientations.
(PDF)

**S1 Table.** **Read counts.**
(XLSX)

**S2 Table.** **Concentrations of total and poly(A)-selected RNAs.**
(DOCX)

**S3 Table.** **Identification of the EHV-1 immediate-early gene through the inhibition of protein synthesis.** In this experiment, we utilized two concentrations of CHX (20 and 100 mg/

ml) and observed the effects at two different incubation durations (6 and 8 hours). Our findings unequivocally indicate that ORF64 is the sole immediate early gene of EHV-1. The numbers refer to the number of the detected reads. Several EHV-1 genes exhibit low-level expression, a phenomenon that can be attributed to transcriptional noise, which becomes more pronounced with reduced CHX concentrations and extended exposure times. (XLSX)

**S4 Table. Previously annotated and novel EHV-1 transcripts.** This table summarizes previously annotated and novel transcripts, along with their TSSs, validated through multiple sequencing approaches, including CAGE, direct cDNA, and direct RNA sequencing. Details of transcript features and splicing events are organized across three sheets: *Sheet A: Previously Published Transcripts* Lists previously annotated transcripts and incorporates their TSSs validated in this study. *Sheet B: Novel Transcripts* Provides details of transcripts newly identified in this study through TSS-TES pairing. *Sheet C: Comparison of LoRTIA and NAGATA for Intron Annotation* Lists introns annotated from dRNA sequencing data, comparing results obtained using LoRTIA and NAGATA. Columns include: • *Gene*: name of the associated gene. • *Transcript name*: identifier of the transcript. • *Transcript category*: type of transcript (e.g., monocistronic, multicistronic, antisense, or non-coding). • *Start and Stop sites (5′- and 3′-ends), and introns*: genomic coordinates of the transcript and intron positions (if spliced). • *Intron donor and acceptor sites:* genomic coordinates of the intron donor and acceptor sites. • *Sequencing validation*: indicates the presence or absence of the transcript's TSS as identified by dRNA-Seq (via NAGATA), CAGE-Seq, and dcDNA-Seq. • *Confirmed putative embedded mRNAs*: highlights previously annotated transcripts that met the 5% relative abundance threshold in CAGE data (compared to the canonical transcript), which was applied to novel putative mRNAs. (XLSX)

**S5 Table. TSS, TES and transcript abundances in the time-resolved dcDNA-Seq datasets and de novo clustering results.** This table consists of four sheets presenting EHV-1 gene-level abundances at multiple infection time points (1–48 hpi) along with their *de novo* clustering results. Each row includes the gene or its canonical TSS or TES, its preassigned kinetic class (IE/E/L/ unknown), and the *de novo* cluster assignment. The columns labeled EHV-1_[Time]_[Rep] indicate viral read count–normalized abundances for each replicate (Rep) at the given time (Time). • *Sheet A*: *TSS-Only*: Contains reads covering the canonical TSS, regardless of TES overlap. • *Sheet B*: *TES-Only*: Contains reads covering the canonical TES, regardless of TSS overlap. • *Sheet C: Transcript Dynamics (TSS+TES):* Contains only reads overlapping both the canonical TSS and TES. • *Sheet D*: *Non-Normalized*: Contains all raw dcDNA-Seq counts, including TSS and TES. Empty fields represent cases where either the TSS or the TES did not overlap. (XLSX)

## Author contributions

**Conceptualization:** Dóra Tombácz, Zsolt Boldogkői.

**Data curation:** Balázs Kakuk, Gábor Torma, Ádám Fülöp, Gábor Gulyás.

**Formal analysis:** Dóra Tombácz, Balázs Kakuk, Gábor Torma, Ádám Fülöp, Ákos Dörmő, Zsolt Boldogkői.

**Funding acquisition:** Dóra Tombácz, Zsolt Boldogkői.

**Investigation:** Dóra Tombácz, Ádám Fülöp, Ákos Dörmő, Gábor Gulyás, Zsolt Csabai.

**Methodology:** Dóra Tombácz, Balázs Kakuk, Gábor Torma, Gábor Gulyás, Zsolt Boldogkői.

**Project administration:** Zsolt Boldogkői.

**Resources:** Zsolt Boldogkői.

**Supervision:** Dóra Tombácz, Zsolt Boldogkői.

**Validation:** Gábor Torma, Ákos Dörmő, Zsolt Csabai.

**Visualization:** Balázs Kakuk, Gábor Torma, Ádám Fülöp, Gábor Gulyás.

**Writing – original draft:** Zsolt Boldogkői.

**Writing – review & editing:** Dóra Tombácz, Balázs Kakuk, Zsolt Boldogkői.

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
