## [Decision Letter · Decision Letter 0]

5 Nov 2024

PONE-D-24-39609Mapping the Temporal Transcriptomic Signature of a Viral Pathogen through CAGE and Nanopore sequencingPLOS ONE

Dear Dr. Boldogkői,

Thank you for submitting your manuscript to PLOS ONE. After careful consideration, we feel that it has merit but does not fully meet PLOS ONE’s publication criteria as it currently stands. Therefore, we invite you to submit a revised version of the manuscript that addresses the points raised during the review process.

We look forward to receiving your revised manuscript.

Kind regards,

Mohamed Shaalan

Academic Editor

PLOS ONE

Journal Requirements:

 'National Research, Development and Innovation Office grant: K 142674 (ZB) and FK 142676 (DT)'

Please state what role the funders took in the study.  If the funders had no role, please state: ''The funders had no role in study design, data collection and analysis, decision to publish, or preparation of the manuscript.'' 

Reviewers' comments:

Reviewer's Responses to Questions

**Comments to the Author**

1. Is the manuscript technically sound, and do the data support the conclusions?

Reviewer #1: Partly

Reviewer #2: Yes

2. Has the statistical analysis been performed appropriately and rigorously? 

Reviewer #1: No

Reviewer #2: Yes

3. Have the authors made all data underlying the findings in their manuscript fully available?

Reviewer #1: No

Reviewer #2: Yes

4. Is the manuscript presented in an intelligible fashion and written in standard English?

Reviewer #1: No

Reviewer #2: Yes

5. Review Comments to the Author

Reviewer #1: In this article, the authors a time-resolved analysis of viral transcript expression during the course of equid alphaherpesvirus 1 (EHV-1) infection including a re-annotation of the EHV-1 transcriptome. For this they combine nanopore sequencing of cDNA (what they denote as direct cDNA sequencing = dcDNA-Seq) and CAGE sequencing to accurately determine transcript starts.

While the underlying experiments appear to be sound as well as the raw data analysis as outlined in methods, there are several major issues with this article:

1) While it is clear why the authors perform CAGE, it remains unclear what the advantage is of now performing dcRNA-seq rather than direct RNA sequencing, which the authors already previously performed. Furthermore, they already published an article performing a re-annotation of the EHV-1 transcriptome (Tombácz, Dóra et al. Heliyon, Volume 9, Issue 7, e17716) but no comparison to this re-annotation is performed. It is thus unclear how their new annotation (the second within a year) compares to the previous one. For instance, they find that “Specifically, 251 transcripts received the highest level of support (***), indicating robust validation by CAGE-Seq. Medium support (**) was found for 47 transcripts, while the lowest level of support (*) was seen in 51 transcripts (Supplementary Table 2).” But it is unclear whether these are novel transcripts detected or previously annotated ones, in particular ones from their previous study.

2) Often the conclusions presented in the article are only substantiated with some references to figures or supplementary tables without any explanation on how they come to this conclusion. This applies e.g. to “Subsequently, we performed long-read dcDNA sequencing. Our findings reinforced that orf64 is the sole IE gene in EHV-1 (Supplementary Table 1).” While in this table ORF64 is the only one with high numbers of reads (which are not normalized to transcript length by-the-way), they find reads for other ORFs at lower levels. So, they need to be clearer regarding why they do not consider the other ORFs IE. Notably, it is never explained which time-points would be considered immediate-early, early or late, making it even more difficult to follow the authors’ logic.

Another example is the kinetic profiling of EHV-1 Transcripts, which is poorly described. In particular, they perform two different types of normalization (one to viral and one to host reads), without analyzing whether these provide different results or why. It also remains unclear whether the grouping of ORFs is based on the IE, E and L annotation or the new clustering they performed and whether both are consistent. They also do not explain what viral read count-normalized TSS-TES dynamics are for which they perform the clustering.

3) Obvious and important analyses are missing, this includes e.g.

- A detailed analysis of the link between TSS and TES sites. They find that there is sometimes a discrepancy between the kinetics of TSS sites and TES sites, and they attribute this to the presence of multicistronic ORFs and alternative TES sites for a gene. However, since they performed long read sequencing they can match TSS to TES sites and explicitly analyze whether this is the explanation or is something else is going on.

- The dynamics of spliced transcripts expression focuses only on the presence of spliced vs. unspliced transcripts but does not take into account the presence of different spliced variants for the same gene. This needs to be analyzed, whether there is just a general splicing trend during infection or whether particular spliced transcripts.

Some other issues:

1) The manuscript is written in a very confusing manner. In particular, the outline of the study is only described at the beginning of the results rather than at the end of the introduction, which makes the methods section difficult to understand. There is also no explanation on why they do not use direct RNA sequencing as in their previous study but need dcDNA-seq here.

2) Upper- and lower case is used inconsistently for ORFs and genes.

3) The noir/NOIR gene/transcript appears to be a novel transcript from their previous study, but that is never explained.

4) The font size in figures is generally very small, making it often almost impossible to discern the ORFs they refer to in the manuscript and at least partially confirm their conclusions. Furthermore, x- and y-axis labels are often missing as well as color legends within the figures. While sometimes this is explained in the caption to the figures, it is not done so consistently and makes it difficult to understand the figures.

5) Their “in-house developed R pipeline” should be made available either as supplement or Zenodo/Github.

6) It is unclear why they explicitly note some R packages but not the “other R-packages from the Bioconductor repository” .

7) I do not understand the point of Figure 6.

Reviewer #2: This is a fine sequencing paper for equine herpesvirus. It is well written and is in line with the large volume of stellar sequencing manuscripts published by this group. I appreciate their work. However, it would be nice to use proteomic to define the alternatively spliced variants of the transcriptome of herpesviruses. So a transcriptome and proteome manuscript would be even better.

6. PLOS authors have the option to publish the peer review history of their article (what does this mean? ). If published, this will include your full peer review and any attached files.

**Do you want your identity to be public for this peer review?** For information about this choice, including consent withdrawal, please see our Privacy Policy .

Reviewer #1: No

Reviewer #2: No

---

## [Author Response · Author response to Decision Letter 1]

5 Jan 2025

Reviewer #1:

1) While it is clear why the authors perform CAGE, it remains unclear what the advantage is of now performing dcRNA-seq rather than direct RNA sequencing, which the authors already previously performed. Furthermore, they already published an article performing a re-annotation of the EHV-1 transcriptome (Tombácz, Dóra et al. Heliyon, Volume 9, Issue 7, e17716) but no comparison to this re-annotation is performed. It is thus unclear how their new annotation (the second within a year) compares to the previous one. For instance, they find that “Specifically, 251 transcripts received the highest level of support (***), indicating robust validation by CAGE-Seq. Medium support (**) was found for 47 transcripts, while the lowest level of support (*) was seen in 51 transcripts (Supplementary Table 2).” But it is unclear whether these are novel transcripts detected or previously annotated ones, in particular ones from their previous study.

We thank the reviewer for the detailed comments and the opportunity to clarify the methodological choices and novel contributions of our study.

Direct RNA sequencing (dRNA-Seq) is valuable for identifying splice sites and transcription end sites (TESs) due to the use of oligodT primers and the low variance in TESs of alphaherpesviruses. However, this technique lacks accurate 5' end information due to 5' truncation caused by motor protein stalling, which limits its reliability in identifying transcription start sites (TSSs). Additionally, dRNA-Seq is prone to various types of errors compared to dcDNA-RNA-Seq (e.g., base-calling issues and read truncation), motivating us to apply a combined approach. Integrating both techniques allows us to achieve more reliable results by leveraging the strengths of each method.

In this study, we employed CAGE-Seq, which is widely accepted for its high resolution in TSS identification, specifically to accurately determine TSSs, complementing the TES and splice site information obtained from dRNA-Seq. While our previous publication focused on annotating canonical transcripts, the current study allowed us to identify additional transcript isoforms and novel TSSs. Here, dcDNA-Seq proved particularly suitable, as it avoids amplification steps and generates full-length reads with true TSSs, thanks to the presence of adapters at the ends of reads (no adapters are used for dRNA-Seq). Thus, dcDNA-Seq provides a more accurate depiction of transcriptional timing.

We appreciate the reviewer’s request for a clearer comparison with our previous annotation. In the revised manuscript, we explicitly distinguish between novel transcripts and those previously detected. While refining the transcriptome annotation is a significant advancement, the primary novelty of our study lies in the comprehensive investigation of temporal transcriptional dynamics, providing new insights into the regulatory complexity of EHV-1. These differences are highlighted in Supplementary Table S2 and the revised text.

2) Often the conclusions presented in the article are only substantiated with some references to figures or supplementary tables without any explanation on how they come to this conclusion. This applies e.g. to “Subsequently, we performed long-read dcDNA sequencing. Our findings reinforced that orf64 is the sole IE gene in EHV-1 (Supplementary Table 1).” While in this table ORF64 is the only one with high numbers of reads (which are not normalized to transcript length by-the-way), they find reads for other ORFs at lower levels. So, they need to be clearer regarding why they do not consider the other ORFs IE. Notably, it is never explained which time-points would be considered immediate-early, early or late, making it even more difficult to follow the authors’ logic. Another example is the kinetic profiling of EHV-1 Transcripts, which is poorly described. In particular, they perform two different types of normalization (one to viral and one to host reads), without analyzing whether these provide different results or why. It also remains unclear whether the grouping of ORFs is based on the IE, E and L annotation or the new clustering they performed and whether both are consistent. They also do not explain what viral read count-normalized TSS-TES dynamics are for which they perform the clustering.

In the revised manuscript, we have added explanations to better substantiate our conclusions.

Regarding ORF64, the CHX treatment identified this gene as the sole immediate-early (IE) gene. We concluded this based on two observations from the data in Supplementary Table S1: (1) only ORF64 transcripts are expressed at high levels, and (2) increasing CHX concentration further amplifies the difference in expression between ORF64 and other genes. The remaining genes are expressed at such low levels that they likely represent transcriptional noise, which occurs because CHX does not fully inhibit protein synthesis. This is a normal phenomenon.

We are uncertain about the reviewer’s suggestion to normalize by transcript length, as we are not aware of a statistically validated method tailored for dcDNA-Seq data. Since dcDNA-Seq is performed without amplification or deliberate fragmentation of native RNA, traditional length-based normalization approaches (e.g., RPKM) are not strictly necessary. Although longer transcripts (e.g., ORF64) may be underrepresented as full-length molecules, we accounted for this by analyzing TSSs, TESs, and transcripts independently. In the revised manuscript, we have also clarified the time points associated with IE, E, and L transcripts as requested by the reviewer.

In the revised manuscript, we have included a more detailed explanation of our de novo clustering methodology for TSSs, TESs, and transcripts as well the evaluation of these results. We introduce a new section, Gene-Level Clustering of Canonical Transcripts, wherein we discuss in detail these results. After careful consideration prompted by the reviewer’s feedback, we decided to use only viral read counts for normalization. Initially, we explored host-read normalization, but this approach skewed the results because many viral genes increase dramatically over time relative to host transcripts, making direct comparisons less transparent. Using viral reads alone allowed us to capture the relative changes in viral gene expression more accurately, as it provided a consistent reference point within the rapidly changing viral transcript population. We also considered traditional normalization methods, such as those implemented in DESeq2. However, many genes exhibited extremely low counts at certain time points (mainly in 1 and 2 hpi samples), leading to large numbers of zeros. DESeq2 and similar algorithms assume that most genes do not change significantly, an assumption that does not hold in our dataset, where substantial portions of the viral genome undergo marked temporal shifts in expression. As a result, applying these methods would not yield reliable normalization factors. Given these constraints, we chose to normalize to total viral read counts and then cluster genes based on their time-resolved expression patterns. This approach allowed us to identify de novo kinetic classes that better reflect the authentic temporal dynamics of EHV-1 transcription. We now explicitly detail these considerations in the revised text, ensuring that readers understand the rationale behind our normalization strategy and the resulting clustering analyses.

3) Obvious and important analyses are missing, this includes e.g.

3a. A detailed analysis of the link between TSS and TES sites. They find that there is sometimes a discrepancy between the kinetics of TSS sites and TES sites, and they attribute this to the presence of multicistronic ORFs and alternative TES sites for a gene. However, since they performed long read sequencing they can match TSS to TES sites and explicitly analyze whether this is the explanation or is something else is going on.

We appreciate the reviewer’s suggestion to analyze the link between TSS and TES sites more explicitly. In this study, we identified transcripts containing TSS, TES, and intron information, enabling detailed mapping of these elements. However, for certain TSS sites, precisely validated transcripts remained elusive, likely due to their low abundance or their presence in very long polygenic transcripts, which are often underrepresented in long-read sequencing datasets. In the revised manuscript, specifically in the new section “Linking TSS and TES Sites,” we provide additional explanations and references to our analyses that match TSSs to TESs. We also discuss how alternative isoforms and multicistronic arrangements can explain observed discrepancies between TSS and TES kinetics, thereby offering a clearer understanding of these complex transcriptional patterns.

3b. The dynamics of spliced transcripts expression focuses only on the presence of spliced vs. unspliced transcripts but does not take into account the presence of different spliced variants for the same gene. This needs to be analyzed, whether there is just a general splicing trend during infection or whether particular spliced transcripts.

We have revised the manuscript to address the expression patterns of different spliced variants within the same gene, rather than only considering spliced vs. non-spliced aggregates. This analysis now shows whether particular spliced transcripts dominate at specific time points, offering a more nuanced view of splicing dynamics during infection.

Some other issues:

1) The manuscript is written in a very confusing manner. In particular, the outline of the study is only described at the beginning of the results rather than at the end of the introduction, which makes the methods section difficult to understand. There is also no explanation on why they do not use direct RNA sequencing as in their previous study but need dcDNA-seq here.

We have thoroughly revised the manuscript to improve clarity and have restructured the introduction to better outline the study’s objectives and methodology.

Regarding the choice of dcDNA-seq over direct RNA sequencing (dRNA-Seq) in this study, we opted for dcDNA-seq as it provides certain advantages in accurately identifying transcription start sites (TSSs) and full-length transcript structures. Unlike dRNA-Seq, which often truncates the 5' end due to motor protein stalling, dcDNA-seq avoids this issue by producing full-length reads with high reliability. This method also eliminates the need for amplification, preserving the original transcript proportions, which is crucial for accurately assessing transcriptional dynamics. Therefore, dcDNA-seq allowed us to achieve more robust and comprehensive results for the goals of this study. In addition, since dRNA-Seq cannot be barcoded, but dcDNA can be, we were able to sequence a total of 27 samples ranging from hpi1 to 48, a total of 9 time points in triplicates.

2) Upper- and lower case is used inconsistently for ORFs and genes.

We have revised the manuscript and now consistently use capital letters (ORF).

3) The noir/NOIR gene/transcript appears to be a novel transcript from their previous study, but that is never explained.

We have revised the manuscript to clarify the NOIR transcripts’ origin and their relationship to our previous findings in the Comparison of replication origin-associated transcripts of EHV-1 and PRV section.

4) The font size in figures is generally very small, making it often almost impossible to discern the ORFs they refer to in the manuscript and at least partially confirm their conclusions. Furthermore, x- and y-axis labels are often missing as well as color legends within the figures. While sometimes this is explained in the caption to the figures, it is not done so consistently and makes it difficult to understand the figures.

We have improved figure quality, increasing font sizes and adding clearer axis labels, legends, and color indicators. This makes it easier to confirm our conclusions visually.

5) Their “in-house developed R pipeline” should be made available either as supplement or Zenodo/Github.

We have made it our complete analysis and code to generate the plots available on github.

6) It is unclear why they explicitly note some R packages but not the “other R-packages from the Bioconductor repository”.

We have clarified our mention of software tools and R packages, providing a consistent explanation of all packages used, including those from Bioconductor.

7) I do not understand the point of Figure 6.

This figure illustrates raw reads, revealing a much more complex transcriptome structure compared to one based solely on annotated transcripts. The main takeaway of the figure is that RNA molecules exhibit extensive transcriptional overlap; which likely impacts gene expression regulation as transcriptional machineries may interfere within these overlapping regions. We have elaborated on this topic in the revised manuscript.

Reviewer #2:

This is a fine sequencing paper for equine herpesvirus. It is well written and is in line with the large volume of stellar sequencing manuscripts published by this group. I appreciate their work. However, it would be nice to use proteomic to define the alternatively spliced variants of the transcriptome of herpesviruses. So a transcriptome and proteome manuscript would be even better.

We thank the reviewer for the valuable comments and constructive feedback on the manuscript. The suggestion to incorporate a proteomic analysis is well-noted, and we fully recognize the importance of such an approach. However, including this analysis would significantly broaden the scope of the current study. Therefore, while this manuscript does not intend to address proteomics, we plan to consider this in a dedicated future study, where the complexity of proteomic data can be thoroughly explored. Thank you once again for the helpful insights, which have contributed to refining the research direction.

---

## [Decision Letter · Decision Letter 1]

29 Jan 2025

PONE-D-24-39609R1Mapping the Temporal Transcriptomic Signature of a Viral Pathogen through CAGE and Nanopore sequencingPLOS ONE

Dear Dr. Boldogkői,

Thank you for submitting your manuscript to PLOS ONE. After careful consideration, we feel that it has merit but does not fully meet PLOS ONE’s publication criteria as it currently stands. Therefore, we invite you to submit a revised version of the manuscript that addresses the points raised during the review process.

We look forward to receiving your revised manuscript.

Kind regards,

Dr. Mohamed Shaalan

Academic Editor

PLOS ONE

Journal Requirements:

Additional Editor Comments :

Kindly address the reviewer suggestions.

Reviewers' comments:

Reviewer's Responses to Questions

**Comments to the Author**

1. If the authors have adequately addressed your comments raised in a previous round of review and you feel that this manuscript is now acceptable for publication, you may indicate that here to bypass the “Comments to the Author” section, enter your conflict of interest statement in the “Confidential to Editor” section, and submit your "Accept" recommendation.

Reviewer #1: (No Response)

Reviewer #2: All comments have been addressed

2. Is the manuscript technically sound, and do the data support the conclusions?

Reviewer #1: Yes

Reviewer #2: Yes

3. Has the statistical analysis been performed appropriately and rigorously? 

Reviewer #1: Yes

Reviewer #2: Yes

4. Have the authors made all data underlying the findings in their manuscript fully available?

Reviewer #1: Yes

Reviewer #2: Yes

5. Is the manuscript presented in an intelligible fashion and written in standard English?

Reviewer #1: Yes

Reviewer #2: Yes

6. Review Comments to the Author

Reviewer #1: The authors addressed most of my previous concerns, however there are some issues, which require more detail/precision in the manuscript.

In the order of appearance in the text:

- "we used an in-house developed R pipeline." -> Here, they should include the link to the github repository. The link in the data availibilty section is not working though.

- "The CAGEfightR [35] package was used to determine TSS positions. The TSS clusters within a 10 nucleotides window were termed identical." Does TSS cluster refer to the TSS positions outputted by CAGEfightR or are these the clusters obtained after merging TSS within 10 nt.

- "Reference transcript counting": It is unclear what is compared against the reference annotation here. The reads from dcRNA-seq? This needs to be clarified.

- nucleotide vs. nt-s: The authors use both terms, but should only use one consistently

- "The clusters were merged with the dcDNA-Seq dataset" -> The TSS clusters?

- ". , which contains transcript identities based on the alignment of 5′-ends" -> I don't understand what this is supposed to mean or refer to. The TSSs or the dcRNA-seq, if the latter why would only the 5'end be aligned. This needs to be rephrased to be unclear

- "Transcripts were reconstructed by pairing validated TSS peaks": what are validated TSS peaks? peaks confirmed by dcRNA-seq in the previous paragraph? But this already merged transcripts (from dcRNA-seq? unclear) with TSS, why do you need to do it again? Also transcripts are from the dcRNA-seq data or where do they come from?

- "their 3′-ends overlapped a known TES" -> known means from the previous annotation?

- "This approach enabled the integration of the CAGE-Seq and dcDNA-Seq datasets to annotate TSSs."-> but you did this already in the previous section? Why did you do this again?

- "Newly assembled transcripts were integrated with our prior annotation"-> how?

- "Validated transcripts"-> when is a transcript considered validated?

- "To further filter TSS transcripts" -> what are TSS transcripts, it has not been defined to which transcripts this term refers to.

- "To identify groups of TSSs, TESs, and transcripts with similar temporal expression patterns, we performed de novo clustering on normalized gene expression data. [..] For gene clustering, .." -> gene clustering aims to identify groups of transcripts with similar temporal expression patterns? If not how are these groups of transcripts identified. This needs to be clarified.

- "Our findings reinforced that ORF64 is the sole IE gene in EHV-1" -> It needs to be explicitly mentions that ORF64 is the only gene with significant expression levels after CHX treatment.

- "Among the examined transcripts, 251 received the highest level of support (***) - indicating robust validation - while 47 had medium support (**) and 51 showed the lowest level of support (*) (see Methods for details)." -> Methods only talks about assigning confidence to TSS. Are you talking about TSS here? You need to be more precise with your terms!

- OriS = Oris? If yes, please use a consistent notation, if not explain.

- Supplementary Figure S1 needs to indicate the location of raRNA, Ori-L and the other genes/mRNAs mentioned in this section, otherwise it cannot be understood.

- noir = NOIR? If yes, please use a consistent notation, if not explain.

- Legend to Supplementary Figure S2 and Supplementary Figure S5 need to state the scale of the y-axis is determined independently for each time-point

- Supplementary Figure S4 should be before Supplementary Figure S3 und referenced in the previous paragraph, which discusses the traditional kinetic clusters

- Figure 7 should be a Supplementary Figure as it is too large for the main manuscript. The text will be unreadable if it is scaled to fit in the main manuscript.

- "By integrating multiple data sources (dcDNA-Seq, dRNA-Seq, and CAGE-Seq) and using different tools - including LoRTIA

for dcDNA-Seq libraries and NAGATA for dRNA-Seq data" -> where is the dRNA-seq data integrated? This did not become clear in the methods (see also my above questions).

- "The R codes used to perform the analysis and generate the plots are available at: https://github.com/Balays/EHV-1-dynamic" -> the link does not exist.

Reviewer #2: This is an improved manuscript and should be accepted without any reservations. I believe your responses to the reviewer's concerns were adequately addressed.

7. PLOS authors have the option to publish the peer review history of their article (what does this mean? ). If published, this will include your full peer review and any attached files.

**Do you want your identity to be public for this peer review?** For information about this choice, including consent withdrawal, please see our Privacy Policy .

Reviewer #1: No

Reviewer #2: No

---

## [Author Response · Author response to Decision Letter 2]

6 Feb 2025

Reviewer #1: The authors addressed most of my previous concerns, however there are some issues, which require more detail/precision in the manuscript.

In the order of appearance in the text:

- "we used an in-house developed R pipeline." -> Here, they should include the link to the github repository. The link in the data availability section is not working though.

The link has been corrected and should work now.

- "The CAGEfightR [35] package was used to determine TSS positions. The TSS clusters within a 10 nucleotides window were termed identical." Does TSS cluster refer to the TSS positions outputted by CAGEfightR or are these the clusters obtained after merging TSS within 10 nt.

TSS clusters are groups of transcription start sites (TSSs) that are identified and merged within a 10-nucleotide window by CAGEfightR, rather than individual raw TSS positions. The software first detects individual CAGE TSSs (CTSSs) and then groups them into clusters based on the specified distance threshold.

- "Reference transcript counting": It is unclear what is compared against the reference annotation here. The reads from dcRNA-seq? This needs to be clarified.

We introduced the term transfrags to facilitate understanding of the workflow. We use transfrags to refer to a list of unique alignments of the dcDNA reads to the viral genome, considering both exons (matches in the alignment) and introns (appearing as Ns at the nucleotide level). Their characteristics and distribution throughout the dcDNA dataset were used as the basis for downstream analysis. We have clarified that reference transcript counting was performed by comparing the list of assembled transfrags (from dcDNA-Seq) to our previously annotated transcript dataset using GFF-compare. Thus, the comparison was not performed using individual sequencing reads but rather by aligning the transfrags to our reference transcript list to quantify the presence of reference transcripts in each sample.

- nucleotide vs. nt-s: The authors use both terms, but should only use one consistently

We have standardized the terminology and now consistently use the term 'nucleotide' throughout the revised manuscript

- "The clusters were merged with the dcDNA-Seq dataset" -> The TSS clusters?

Yes, this refers to the TSS clusters obtained from CAGEfightR. For clarity, we have explicitly stated 'TSS clusters' in the revised text.

- ". , which contains transcript identities based on the alignment of 5′-ends" -> I don't understand what this is supposed to mean or refer to. The TSSs or the dcRNA-seq, if the latter why would only the 5'end be aligned. This needs to be rephrased to be unclear

This refers to the TSS clusters obtained from CAGEfightR and the transfrags (derived from the dcDNA data). It means that the two datasets were merged based on the genomic position (start, end, and strand) of the former and the 5′-end position (and strand) of the latter. This has been made clearer in the revised article by explicitly using the term 'transfrags'.

- "Transcripts were reconstructed by pairing validated TSS peaks": what are validated TSS peaks? peaks confirmed by dcRNA-seq in the previous paragraph? But this already merged transcripts (from dcRNA-seq? unclear) with TSS, why do you need to do it again? Also transcripts are from the dcRNA-seq data or where do they come from?

CAGE TSS clusters were sometimes broad, exceeding 150 nucleotides in certain cases. To enhance precision, we refined these TSSs using the 5′-end counts of transfrags derived from dcDNA-Seq. For each TSS cluster, peak analysis was performed to identify validated TSS peaks, which differed from the original CAGE TSS clusters but were supported by both CAGE-Seq and dcDNA-Seq data. This refinement ensured that only high-confidence TSSs were used for novel transcript reconstruction. While some transfrags could not be assigned to reference transcripts from our previous study, their strong TSS signals in both dcDNA-Seq and CAGE-Seq indicated that they likely represented previously unannotated transcripts. The full workflow has been clarified in the Methods section, and the correct terminology is dcDNA-Seq, not dcRNA-Seq.

- "their 3′-ends overlapped a known TES" -> known means from the previous annotation?

Yes, 'known' refers to TESs from our previous transcriptome annotation. This has been explicitly clarified in the revised text.

- "This approach enabled the integration of the CAGE-Seq and dcDNA-Seq datasets to annotate TSSs."-> but you did this already in the previous section? Why did you do this again?

This referred to the previous section. We have clarified this distinction in the text.

- "Newly assembled transcripts were integrated with our prior annotation"-> how?

Using the gff-compare output, we analyzed the relationship between each novel transcript and the reference list. Novel transcripts that met the validation criteria - TSS-TES pairing, expression threshold, and support from multiple sequencing methods - were incorporated into the updated annotation. Additionally, we manually reviewed the newly identified transcripts and compared them to our previous transcript list.

- "Validated transcripts"-> when is a transcript considered validated?

A novel transcript (not found in the reference list within the ±10 nucleotide wobble) was considered validated if it met all of the following criteria:

- Supported by at least three independent dcDNA-Seq reads.

- The TSS was within ±10 nt of a validated TSS peak (within a CAGE-Seq TSS cluster).

- The TES overlapped a known TES within ±10 nt.

- Correct 5′- and 3′-adapter sequences were detected.

This definition has been explicitly included in the Methods section.

- "To further filter TSS transcripts" -> what are TSS transcripts, it has not been defined to which transcripts this term refers to.

This was a typographical error; we were referring to 'To further filter the TSS of transcripts …'. This indicates that we applied stricter criteria when accepting 5′-truncated transcripts containing 5′-truncated ORFs to eliminate sequencing artifacts. These artifacts occurred when the reverse transcription step in the dcDNA protocol was incomplete. The error has been corrected.

- "To identify groups of TSSs, TESs, and transcripts with similar temporal expression patterns, we performed de novo clustering on normalized gene expression data. [..] For gene clustering, .." -> gene clustering aims to identify groups of transcripts with similar temporal expression patterns? If not how are these groups of transcripts identified. This needs to be clarified.

Yes, transcripts were grouped into clusters based on their temporal expression profiles. The terms gene expression and transcript expression were used interchangeably, but we have clarified that this analysis specifically refers to transcript expression, as only canonical transcripts were counted and clustered. Unlike gene-level clustering, which includes all transcript isoforms, this approach eliminates ambiguity in assigning isoforms to specific genes. Differential transcript expression, including isoform-specific patterns, was analyzed separately in the sections 'Dynamics of transcriptional isoform switching in selected genes' and 'Dynamics of spliced transcript expression'. These distinctions have now been explicitly clarified in the Methods section.

- "Our findings reinforced that ORF64 is the sole IE gene in EHV-1" -> It needs to be explicitly mentions that ORF64 is the only gene with significant expression levels after CHX treatment.

We have revised the manuscript to explicitly state that ORF64 is the only gene with significant expression levels following CHX treatment. Statistical analyses—including t-tests, Z-score analysis, coefficient of variation (CV), and interquartile range (IQR) outlier detection—confirm that ORF64 is the sole viral gene significantly expressed under CHX treatment, while other detected transcripts likely represent background noise. This finding reinforces ORF64’s classification as the only true immediate-early (IE) gene in EHV-1.

- "Among the examined transcripts, 251 received the highest level of support (***) - indicating robust validation - while 47 had medium support (**) and 51 showed the lowest level of support (*) (see Methods for details)." -> Methods only talks about assigning confidence to TSS. Are you talking about TSS here? You need to be more precise with your terms!

Here, we refer to previously published transcripts - specifically, how many received certain confidence categories in our CAGE-based TSS analysis. As described in the Methods (TSS Clusters Validation), the ‘’, ‘**’, and ‘’ levels apply exclusively to TSS clusters identified via CAGE-Seq. We then merged these clusters with the dcDNA-Seq transfrags, to which reference transcripts had already been assigned by gff-compare based on their 5′-position overlaps. This clarification has been added to the Transcript Merging and TSS Refinement section. The reported numbers - 251 (***), 47 (**), and 51 (*) - represent reference (previously annotated) transcripts that inherited TSS confidence ratings from CAGE-Seq. However, if a transfrag did not align with any known reference transcript but exhibited a strong CAGE signal and a correct dcDNA 5′-adapter signal, we assembled a novel transcript from that transfrag. This process is now detailed in the Transcript Assembly and Validation section.

- OriS = Oris? If yes, please use a consistent notation, if not explain.

OriL and OriS are the correct terms, and we have updated the article accordingly.

- Supplementary Figure S1 needs to indicate the location of raRNA, Ori-L and the other genes/mRNAs mentioned in this section, otherwise it cannot be understood.

In Supplementary Figure S1, the aligned nucleotide sequences of the NOIR (A) and CTO (B) genes in the three viral strains are shown along with their orientation. Since this represents a multiple sequence alignment of genes, including surrounding genomic regions and/or transcripts was not feasible. To improve clarity, we have created two additional figures. Supplementary Figure S2 provides a high-resolution view of the EHV-1 replication origin region, while Supplementary Figure S3 presents the same for PRV. These figures clearly depict the locations of the NOIR and CTO genes, their corresponding transcripts, and the replication origins within the given region. In these figures, molecules overlapping the replication origin are shown in green, those transcribed from the forward strand are in red, and those transcribed from the reverse strand are also in green.

- noir = NOIR? If yes, please use a consistent notation, if not explain.

Any instance of 'noir' has been omitted; we now use 'NOIR' consistently throughout the manuscript.

- Legend to Supplementary Figure S2 and Supplementary Figure S5 need to state the scale of the y-axis is determined independently for each time-point

We have revised the legends for both Supplementary Figure S2 (now S4) and Supplementary Figure S5 (now S6) to explicitly state that each facet’s y-axis is scaled independently (using ggplot’s scale = "free_y"). This ensures that the y-axis range adapts to the data in each time-point facet, allowing for clearer visualization of expression profiles.

- Supplementary Figure S4 should be before Supplementary Figure S3 und referenced in the previous paragraph, which discusses the traditional kinetic clusters

The reviewer is correct once again; we have reordered Supplementary Figures S3 (now S5) and S4 (now S6). Additionally, following the same logic, we also reordered Supplementary Figures S6 (now S8) and S7 (now S9).

- Figure 7 should be a Supplementary Figure as it is too large for the main manuscript. The text will be unreadable if it is scaled to fit in the main manuscript.

We agree. Figure S7 has been renumbered as Supplementary Figure S13 to improve the visibility of annotations and other text.

- "By integrating multiple data sources (dcDNA-Seq, dRNA-Seq, and CAGE-Seq) and using different tools - including LoRTIA for dcDNA-Seq libraries and NAGATA for dRNA-Seq data" -> where is the dRNA-seq data integrated? This did not become clear in the methods (see also my above questions).

We utilized our previous dRNA-Seq dataset and the NAGATA software to (1) validate introns and (2) confirm the presence of newly identified TSSs, with a specific focus on 5′-truncated ORF-carrying 'putative transcripts.' This approach was primarily aimed at minimizing the risk of annotating sequencing artifacts caused by enzyme stalling during reverse transcription, a known issue in dcDNA-Seq. These validation steps were undertaken to complement and verify both the sequencing methods used in this study (dcDNA-Seq and CAGE-Seq) and the bioinformatic approach.

In the Methods section (see Filtering 5′-Truncated ORF-Carrying Transcripts and Transcript Assembly and Validation), we detail how we reanalyzed our earlier dRNA-Seq data with NAGATA software to ensure that 5′-truncated ORF isoforms—and the introns identified in dcDNA-Seq—were also supported by dRNA reads. Specifically:

Intron Validation: Introns detected in dcDNA-Seq were accepted only if they also appeared in our previous dRNA-Seq dataset.

TSS Cross-Checking: Any novel TSS identified in dcDNA-Seq and CAGE-Seq had to be detectable within a 25-nt window in the dRNA-Seq-NAGATA output to confirm its authenticity.

This integrated approach helped eliminate potential artifacts and reinforced our confidence in newly discovered TSSs and alternative isoforms."

- "The R codes used to perform the analysis and generate the plots are available at: https://github.com/Balays/EHV-1-dynamic" -> the link does not exist.

The provided link https://github.com/Balays/EHV-1-dynamic is now functional, and the scripts used to generate the analysis and figures are accessible.

---

## [Decision Letter · Decision Letter 2]

19 Feb 2025

Mapping the Temporal Transcriptomic Signature of a Viral Pathogen through CAGE and Nanopore sequencing

PONE-D-24-39609R2

Dear Dr. Boldogkői,

We’re pleased to inform you that your manuscript has been judged scientifically suitable for publication and will be formally accepted for publication once it meets all outstanding technical requirements.

Kind regards,

Dr. Mohamed Shaalan

Academic Editor

PLOS ONE

**Comments to the Author**

Reviewer #1: All comments have been addressed

---

## [Editor Report · Acceptance letter]

PONE-D-24-39609R2

PLOS ONE

Dear Dr. Boldogkői,

I'm pleased to inform you that your manuscript has been deemed suitable for publication in PLOS ONE. Congratulations! Your manuscript is now being handed over to our production team.

Kind regards,

on behalf of

Dr. Mohamed Shaalan

Academic Editor

PLOS ONE